# TV-Rec: Time-Variant Convolutional Filter for Sequential Recommendation

**Yehjin Shin    Jeongwhan Choi    Seojin Kim    Noseong Park**[*]
KAIST
{yehjin.shin, jeongwhan.choi, seojinkim, noseong}@kaist.ac.kr

## Abstract

Recently, convolutional filters have been increasingly adopted in sequential recommendation for their ability to capture local sequential patterns. However, most of these models complement convolutional filters with self-attention. This is because convolutional filters alone, generally fixed filters, struggle to capture global interactions necessary for accurate recommendation. We propose **T**ime-**V**ariant Convolutional Filters for Sequential **Rec**ommendation (TV-Rec), a model inspired by graph signal processing, where time-variant graph filters capture position-dependent temporal variations in user sequences. By replacing both fixed kernels and self-attention with time-variant filters, TV-Rec achieves higher expressive power and better captures complex interaction patterns in user behavior. This design not only eliminates the need for self-attention but also reduces computation while accelerating inference. Extensive experiments on six public benchmarks show that TV-Rec outperforms state-of-the-art baselines by an average of 7.49%.

## 1  Introduction

Recommender systems have become essential for guiding users through vast amounts of content by providing personalized information based on users' historical interactions [41, 13, 1, 19, 15, 2, 3, 8, 16, 21]. Considering that preferences evolve over time, sequential recommendation (SR) has become widely used for capturing dynamic preferences using sequential patterns in users' interactions. Various approaches have been developed to more accurately capture users' dynamic sequential patterns, including architectures such as Markov chains [25], RNNs [14], and GNNs [36]. Transformers [32], a powerful architecture in NLP, have been widely adopted as encoders for many SR models, highlighting their ability to model long-term dependencies in data [18, 30, 24, 6].

Despite their modeling power, self-attention mechanisms have a fundamental limitation: they lack an inductive bias toward sequential structure. While position embeddings provide absolute position information, self-attention still treats all positions pairwise without any inherent bias toward local proximity. As a result, it makes it difficult to model fine-grained, localized user behavior patterns [29]. This limitation has led to the development of hybrid models that integrate convolutional layers to introduce locality

Table 1: Comparison of existing methods based on three points: i) convolutional filter type, ii) inference efficiency, and iii) recommendation performance. The double tick mark indicates better performance compared to a single tick mark.

| Model | Convolutional Filter | Self-Attention | Inference Efficiency | Rec. Performance |
|---|---|---|---|---|
| SASRec [18] | ✗ | ✓ | ✓ | ✓ |
| BERT4Rec [30] | ✗ | ✓ | ✓ | ✓ |
| Caser [31] | Fixed | ✗ | ✗ | ✓ |
| NextItNet [42] | Fixed | ✗ | ✗ | ✓ |
| FMLPRec [47] | Fixed | ✗ | ✓ | ✓ |
| AdaMCT [17] | Fixed | ✓ | ✗ | ✓✓ |
| BSARec [29] | Fixed | ✓ | ✗ | ✓✓ |
| TV-Rec (Ours) | Time-Variant | ✗ | ✓ | ✓✓ |

---

[*]Corresponding author.

bias. For instance, AdaMCT [17] utilizes self-attention and 1D convolution together to capture both long-term and short-term user preferences, while BSARec [29] addresses the limitations of self-attention by applying convolution using the Fourier Transform.

At the same time, convolution-only models have been explored for their effectiveness in capturing local sequential dependencies, which are particularly valuable in SR. As shown in Table 1, models such as Caser [31], NextItNet [42], and FMLPRec [47] use fixed convolutional filters to detect patterns in user sequences. These filters focus on users' recent behavior, which is advantageous in recommending users' next items. However, their fixed nature limits their adaptability: the same filter is applied uniformly across positions, making it difficult to capture temporally evolving or position-specific semantics. While convolutional filters solve this issue by using multiple fixed filters to capture various patterns, these filters remain static and cannot adapt to the specific context or temporal variations at each position in the sequence. This limits their expressiveness in modeling evolving user preferences, especially when user interests shift rapidly over time.

This reveals a fundamental trade-off: convolution excels at modeling local patterns but lacks flexibility, whereas self-attention is expressive but inefficient and insensitive to locality. To bridge this gap, we propose a new architecture called **T**ime-**V**ariant Convolutional Filters for Sequential **Rec**ommendation (TV-Rec), which captures both local and global patterns while achieving greater efficiency than existing hybrid models such as AdaMCT and BSARec. We design time-variant convolutional filters to effectively capture temporal variations and emphasize the most relevant elements at each time step. Our key finding is that existing complicated models based on self-attention and convolutional filters can be replaced with our time-variant convolutional filter.

Inspired by graph signal processing (GSP), we reinterpret SR as a line graph and, instead of using fixed convolution filters, apply time-variant graph filters, analogous to node-variant filters in the graph domain. These filters enable us to effectively adapt weights across the sequence, eliminating the need for positional embeddings while directly encoding temporal signals. Moreover, the time-variant filters act as linear operators, resulting in faster inference with lower complexity.

To evaluate the effectiveness of TV-Rec, we conduct extensive experiments on 6 benchmark datasets. Our results indicate that TV-Rec consistently outperforms state-of-the-art baseline methods. Additionally, we perform a series of experiments comparing the theoretical and practical complexity of TV-Rec with other recent hybrid baselines that combine convolution and self-attention. These experiments demonstrate improved recommendation accuracy and enhanced generalization capabilities. The contributions of this work are as follows:

- We propose **T**ime-**V**ariant Convolutional Filters for Sequential **Rec**ommendation (TV-Rec), using time-variant convolutional filters to capture temporal dynamics and user behavior patterns more effectively (Sec. 3).

- We show that TV-Rec provides more expressive generalization (Sec. 3.4) and effectively captures both long-term preferences and recent interests via filters designed as functions of time (Sec. 4.5).

- We conduct extensive experiments on 6 benchmark datasets, and the results demonstrate that TV-Rec outperforms state-of-the-art baseline methods by an average of 7.49% (Sec. 4.2), while achieving the optimal balance between accuracy and efficiency (Sec. 4.6).

## 2   Preliminaries

In this section, we present the problem statement and the notations used in this paper. We then discuss GSP and the node-variant graph filter, which are core components of TV-Rec.

### 2.1   Problem Statement

SR aims to model user behavior sequences based on implicit feedback to predict and recommend the user's next interaction. Assume that we have a set of users $\mathcal{U}$ and a set of items $\mathcal{V}$, where $|\mathcal{U}|$ and $|\mathcal{V}|$ denote the total numbers of users and items, respectively. The interacted items of each user $u \in \mathcal{U}$ can be chronologically ordered into a sequence $\mathcal{S}^{(u)} = [v_1^{(u)}, v_2^{(u)}, \ldots, v_{|\mathcal{S}^{(u)}|}^{(u)}]$, where $v_i^{(u)}$ represents the $i$-th item in the sequence of user $u$. For simplicity, the superscript $(u)$, which indicates the user, will

be omitted henceforth. Therefore, the goal is to predict $p(v_{|\mathcal{S}|+1} = v \mid \mathcal{S})$ and recommend a Top-$r$ list of items as potential next interactions in the sequence.

## 2.2 Graph Signal Processing (GSP)

Our method, TV-Rec, incorporates key concepts from GSP. GSP analyzes signals on graphs, with graph filtering as a core operation that emphasizes or suppresses specific frequency components of the signal. Given a shift operator $\mathbf{S} \in \mathbb{R}^{N \times N}$, which can be an adjacency matrix or a Laplacian matrix, a graph filter $\mathbf{G}$ is defined as a polynomial of $\mathbf{S}$:

$$\mathbf{y} = \mathbf{G}\mathbf{x} = \sum_{k=0}^{K} h_k \mathbf{S}^k \mathbf{x}, \tag{1}$$

where $\mathbf{x} \in \mathbb{R}^N$ is a graph signal, $h_k$ are filter taps and $K$ is the order of the filter.

This operation can be interpreted in the frequency domain using the graph Fourier transform (GFT), which enables the decomposition of graph signals into different frequency components. Given a shift operator $\mathbf{S}$, the GFT is defined using its eigen-decomposition[2]:

$$\mathbf{S} = \mathbf{U} \, \mathtt{diag}(\boldsymbol{\lambda}) \mathbf{U}^\top, \tag{2}$$

where $\mathbf{U}$ is the matrix of eigenvectors, $\boldsymbol{\lambda}$ is the vector of eigenvalues, and $\mathtt{diag}(\cdot)$ indicates constructing a diagonal matrix from a vector. The GFT of a graph signal $\mathbf{x}$ is $\tilde{\mathbf{x}} = \mathbf{U}^\top x$, and the inverse GFT is $\mathbf{x} = \mathbf{U}\tilde{\mathbf{x}}$. In the frequency domain, the graph filter $\mathbf{G}$ acts on the transformed signal $\tilde{\mathbf{x}}$:

$$\tilde{\mathbf{y}} = \tilde{\mathbf{G}}\tilde{\mathbf{x}} = \Big( \sum_{k=0}^{K} h_k \, \mathtt{diag}(\boldsymbol{\lambda})^k \Big)\tilde{\mathbf{x}}, \tag{3}$$

where $\tilde{\mathbf{G}}$ forms a diagonal matrix, allowing the calculation between the graph filter and the signal to be element-wise multiplication. The filtered signal in the time domain can be obtained by applying inverse GFT, $\mathbf{y} = \mathbf{U}\tilde{\mathbf{y}}$.

In SR, GSP can be utilized to model user behavior sequences as graph signals by representing items as nodes in a line graph. By applying graph filters, we can capture the complex dependencies between items and enhance the predictive performance of SR. Graph convolution in this setting aggregates information from neighboring items in the sequence, which are close in time, allowing the model to learn both local and global patterns in user interactions.

## 2.3 Node-Variant Graph Filter

We focus on node-variant graph filters [28, 7], that apply distinct filter taps at each node position. As shown in Fig. 1, conventional graph convolutional filters introduced above use a scalar as the filter tap ($h_k$), whereas node-variant graph filters $\mathbf{G}_{nv}$ use a vector ($\mathbf{h}_k$) for the filter tap, as follows:

$$\mathbf{y} = \mathbf{G}_{nv}\mathbf{x} = \Big( \sum_{k=0}^{K} \mathtt{diag}(\mathbf{h}_k) \mathbf{S}^k \Big)\mathbf{x}, \tag{4}$$

where $\mathtt{diag}(\cdot)$ indicates constructing a diagonal matrix from a vector, meaning that every node has a different filter tap $\mathbf{h}_k = [h_k^{(1)}, h_k^{(2)}, \cdots, h_k^{(N)}]$. The set of filter taps can be represented in matrix form as $\mathbf{H} \in \mathbb{C}^{N \times (K+1)}$, where the $k$-th column is $\mathbf{h}_k$.

While conventional graph convolutional filters are calculated by element-wise multiplication of the frequency responses of the graph signal and the filter, the frequency response of a node-variant graph filter is given as follows:

$$\tilde{\mathbf{y}} = \tilde{\mathbf{G}}_{nv}\tilde{\mathbf{x}} = \mathbf{U}^\top(\mathbf{U} \circ (\mathbf{H}\boldsymbol{\Lambda}^\top))\tilde{\mathbf{x}}, \tag{5}$$

---

[2]In the general case, the GFT considers the Jordan decomposition $\mathbf{S} = \mathbf{V}\mathbf{J}\mathbf{V}^{-1}$, but we assume that $\mathbf{S}$ is a diagonalizable matrix, so the Jordan decomposition is equivalent to the eigen-decomposition.

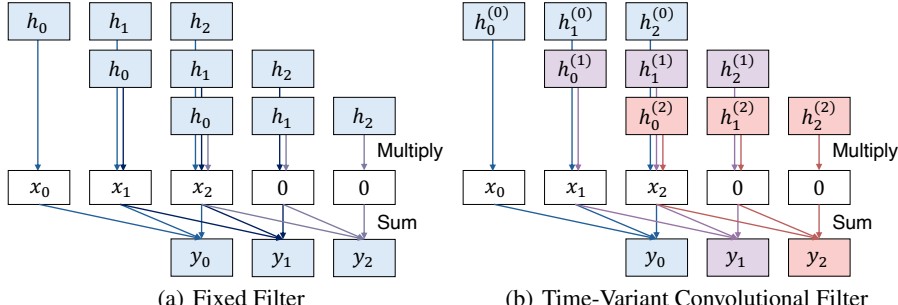

(a) Fixed Filter     (b) Time-Variant Convolutional Filter

Figure 1: Comparison of a fixed filter in (a) and a time-variant convolutional filter in (b) under our line graph expression of a sequence of signals $x_i$, with $K = 2$ and $N = 3$. The output $y_j$, i.e., the filtered signal at index $j$, is produced by summing the filtered results. Arrow colors show each filter's contribution to the output $y_j$, while different $h_i$ box colors represent different filters. In the fixed filter case in (a), the same filter $h_i$ is applied to every node, while the time-variant convolutional filter in (b) allows each node to have its own filter.

where $\mathbf{\Lambda} \in \mathbb{C}^{N \times (K+1)}$ is a Vandermonde matrix[3] given by $\mathbf{\Lambda}_{ik} = \lambda_i^{k-1}$ and $\circ$ denotes the element-wise product of matrices. The proof for the frequency response of the node-variant graph filter can be found in Appendix A.

Node-variant graph filters provide a flexible, general approach to creating operators while preserving local implementation, effectively adapting to changing user preferences. In our line graph context, where each node represents a distinct time point, these filters function equivalently to *time-variant convolutional filters*. For consistency, we will use this term in the following sections.

## 3    Proposed Method

In this section, we present the design of TV-Rec. As shown in Fig. 2, TV-Rec consists of 3 modules: embedding layer, time-variant encoder, and prediction layer.

### 3.1    Embedding Layer

We first convert a user's historical interaction sequence $\mathcal{S}$ to a fixed length $N$. If $|\mathcal{S}| \geq N$, we truncate the sequence keeping the most recent $N$ items, and if $|\mathcal{S}| < N$, we pad the sequence with zeros at the beginning. This process results in a sequence of

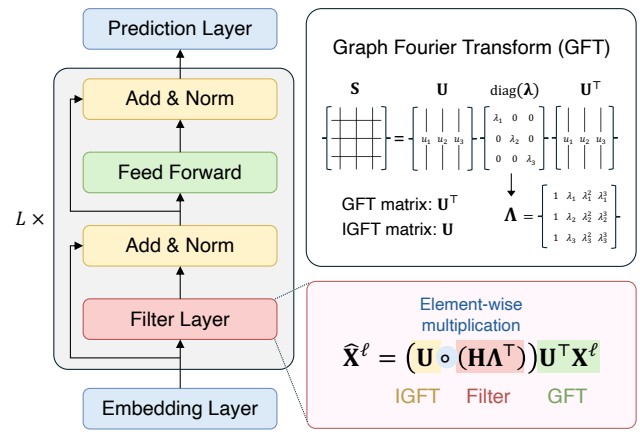

Figure 2: Architecture of our proposed TV-Rec.

length $N$, denoted as $s = (s_1, s_2, \cdots, s_N)$. Using the item embedding matrix $\mathbf{E} \in \mathbb{R}^{|\mathcal{V}| \times D}$ where $D$ is the latent dimension size, we then apply a look-up operation to obtain the embedding representation of the user sequence, followed by layer normalization and dropout. This process produces the final embedding of the user sequence $\mathbf{X}^0$, serving as the input for the time-variant encoder:

$$\mathbf{X}^0 = \text{Dropout}(\text{LayerNorm}([\mathbf{E}_{s_1}, \mathbf{E}_{s_2}, \cdots, \mathbf{E}_{s_N}]^\top)), \tag{6}$$

where $\mathbf{E}_v$ denotes the embedding of item $v$ from $\mathbf{E}$. Note that positional embedding is not necessary due to the benefits of applying our time-variant convolutional filters.

---

[3]A Vandermonde matrix has rows formed by the powers of a set of values, with each element in the $i$-th row and $j$-th column given by $x_i^{j-1}$.

## 3.2 Time-Variant Encoder

We build our item encoder by stacking $L$ time-variant encoding blocks, each containing a filter layer, a feed-forward network, and a residual connection applied after both.

**Filter Layer.** In the $\ell$-th filter layer, with $\mathbf{X}^\ell$ as the input, we perform a filtering operation, then apply a residual connection and layer normalization. As shown in Fig. 2, we first transform $\mathbf{X}^\ell$ into the frequency domain as $\widetilde{\mathbf{X}}^\ell = \mathbf{U}^\top \mathbf{X}^\ell$, where $\mathbf{U}$ denotes the GFT matrix derived from a padded directed cyclic graph (DCG), which we adopt in place of a line graph to ensure diagonalizability and enable spectral filtering (see Appendix B for formal justification). Then, we calculate the time-variant convolutional filter using the filter tap $\mathbf{H} \in \mathbb{C}^{N \times (K+1)}$ and the Vandermonde matrix $\mathbf{\Lambda} \in \mathbb{R}^{N \times (K+1)}$:

$$\widehat{\mathbf{X}}^\ell = \mathbf{G}_{nv}\widetilde{\mathbf{X}} = \left(\mathbf{U} \circ (\mathbf{H}\mathbf{\Lambda}^\top)\right)\widetilde{\mathbf{X}}^\ell = \left(\mathbf{U} \circ (\mathbf{H}\mathbf{\Lambda}^\top)\right)\mathbf{U}^\top \mathbf{X}^\ell \tag{7}$$

where $\circ$ indicates element-wise multiplication. Note that the inverse GFT matrix $\mathbf{U}$ is multiplied with the filter earlier than it is with the frequency response of the signal $\widetilde{\mathbf{X}}^\ell$. To enhance expressive power, we construct the filter matrix $\mathbf{H}$ as follows:

$$\mathbf{H} = \mathbf{C}\bar{\mathbf{B}} = \mathbf{C}\left(\frac{\mathbf{B}}{\|\mathbf{B}\|_2}\right), \tag{8}$$

where $\mathbf{C} \in \mathbb{R}^{N \times m}$ is the coefficient matrix that generates position-specific filters, and $\bar{\mathbf{B}} \in \mathbb{C}^{m \times (K+1)}$ is the normalized basis matrix. The parameter $m$ determines the number of basis vectors. Since each node corresponds to a position in the sequence, $\mathbf{C}$ can be considered as a function of time. For numerical stability, we normalize $\mathbf{B}$ using the L2 norm along each row.

After Eq. (7), we use a residual connection with dropout and layer normalization to prevent overfitting:

$$\mathbf{F}^\ell = \text{LayerNorm}(\mathbf{X}^\ell + \text{Dropout}(\widehat{\mathbf{X}}^\ell)). \tag{9}$$

**Feed Forward Layer.** After the filter layer, we employ a feed-forward network for non-linearity:

$$\widehat{\mathbf{F}}^\ell = \text{FFN}(\mathbf{F}^\ell) = (\text{GELU}(\mathbf{F}^\ell \mathbf{W}_1^\ell + \mathbf{b}_1^\ell))\mathbf{W}_2^\ell + \mathbf{b}_2^\ell, \tag{10}$$

where $\mathbf{W}_1^\ell, \mathbf{W}_2^\ell \in \mathbb{R}^{D \times D}$, and $\mathbf{b}_1^\ell, \mathbf{b}_2^\ell \in \mathbb{R}^{D \times D}$ are learnable parameters. As in Eq. (9), we apply a dropout layer, residual connections, and layer normalization to get the output of $\ell$'s layer as follows:

$$\mathbf{X}^{\ell+1} = \text{LayerNorm}(\mathbf{F}^\ell + \text{Dropout}(\widehat{\mathbf{F}}^\ell)). \tag{11}$$

## 3.3 Prediction Layer and Training

**Prediction Layer.** After processing through $L$ time-variant encoding blocks, we compute the user's preference score for each item in the entire item set $\mathcal{V}$ as follows:

$$\hat{y}_v = p(v_{|\mathcal{S}|+1} = v|\mathcal{S}) = \mathbf{E}_v^\top \mathbf{X}_N^L, \tag{12}$$

where $\mathbf{E}_v$ is the embedding of item $v$ and $\mathbf{X}_N^L$ is the final sequence representation.

**Model Training.** Similar to other studies [17, 29, 24], we optimize our model using cross-entropy loss $\mathcal{L}_{\text{ce}}$ with an orthogonal regularization term $\mathcal{L}_{\text{ortho}}$ on the basis matrix $\mathbf{B}$ used in Eq. (8):

$$\mathcal{L} = \underbrace{-\log\frac{\exp(\hat{y}_g)}{\sum_{v \in \mathcal{V}}\exp(\hat{y}_v)}}_{\mathcal{L}_{\text{ce}}} + \alpha \cdot \underbrace{\left(\left\|\mathbf{B}_{\text{real}}\mathbf{B}_{\text{real}}^\top - \mathbf{I}\right\|_F^2 + \left\|\mathbf{B}_{\text{imag}}\mathbf{B}_{\text{imag}}^\top - \mathbf{I}\right\|_F^2\right)}_{\mathcal{L}_{\text{ortho}}}, \tag{13}$$

where $g$ is the ground-truth item, $\mathbf{B}_{\text{real}}$ and $\mathbf{B}_{\text{imag}}$ denote the real and imaginary components of $\mathbf{B}$, $\alpha$ controls the regularization strength, $\mathbf{I}$ denotes the identity matrix, and $F$ denotes the Frobenius norm.

### 3.4 Discussion

**Relations to Other Methods.** From a graph filtering perspective, several existing SR methods can be viewed as special cases within our time-variant filter. In particular, 1D CNN in AdaMCT corresponds to a fixed graph convolutional filter, $\mathbf{G}$ in Eq. (1), where $K$ is the kernel size. Similarly, FMLPRec and BSARec apply the discrete Fourier transform (DFT), mathematically equivalent to the GFT of a DCG [27, 26, 29], representable as $\tilde{\mathbf{G}}$ in Eq. (3). Our time-variant filter is a more general method that can be reduced to these approaches as special cases when the filters are fixed. By contrast, Transformer-based studies that interpret self-attention via graph filtering [4, 34] differ from our approach, as TV-Rec reformulates SR as graph signal filtering without relying on self-attention.

**Comparison to GNN-based Methods.** TV-Rec relates to GNN-based methods that model item dependencies through item-transition graphs [35, 37, 45], but differs in two key aspects. First, TV-Rec defines sequence positions rather than items as graph nodes. This enables the model to capture fine-grained temporal and positional dependencies that are often ignored when identical items are merged in a graph. Second, while GNN-based models rely on iterative message passing, TV-Rec employs time-variant graph convolutional filters directly in the spectral domain without recursive propagation, yielding a more efficient representation of temporal dynamics.

**Why We Need Time-Variant Graph Filters?** In Fig. 1, a fixed filter applies the same weights in the sequence, emphasizing recent items. However, it also makes it difficult to capture specific patterns at different stages. For instance, while the patterns at the end of the sequence may highlight recent items, the patterns at the beginning can provide crucial insights into the user's overall preferences. As a result, the fixed filter may lose valuable information, particularly when attempting to understand early-stage patterns. In contrast, our time-variant filter uses different filters for each position, allowing the model to capture both recent items and long-term preferences.

**Why Positional Encoding is Unnecessary?** Unlike Transformer-based models that require explicit positional encodings, TV-Rec naturally encodes positional information via the spectral properties of the graph. Since TV-Rec constructs a DCG and applies the GFT, it shares the same frequency components as sinusoidal encodings (see Appendix C). Furthermore, the time-variant graph filter acts as a position-specific operation, dynamically modulating frequency components without requiring additional embeddings. In Sec. 4.3, we confirm that adding positional embeddings does not offer a performance benefit.

**Time Complexity.** Assume that $n$ is the length of the input sequence and $d$ is the dimension of each input vector. The time complexity of self-attention is $O(nd^2 + n^2d)$, where $O(nd^2)$ is for computing the key, query, and value matrices, and $O(n^2d)$ is for calculating the attention scores and applying them to the value matrix. The time complexity of our time-variant convolutional filter is $O(n^2m + n^2d)$, where $O(n^2m)$ is for computing the filter tap $\mathbf{H}$, and $O(n^2d)$ is for applying GFT to the input signal and multiplying it with the filter tap. Since $m \leq n$, this can be simplified to $O(n^3 + n^2d)$. The difference in complexity between self-attention and the time-variant graph filter depends on the relative sizes of $n$ and $d$, as it determines which term dominates. It is worth noting that the time-variant convolutional filter is a linear operator, so $\mathbf{G}_{nv}$ does not need to be computed for every inference, and can be precomputed after training, resulting in a time complexity of $O(n^2d)$.

## 4 Experiments

### 4.1 Experimental Setup

We evaluate TV-Rec on 6 benchmark datasets for SR, following the preprocessing procedures in [47, 46]. For standard experiments, we set the maximum sequence length $N$ to 50. Additionally, to examine performance on long-range dependencies, we conduct experiments on ML-1M and Foursquare with longer average interactions, setting $N$ to 200. For evaluation, we use standard Top-$r$ metrics (HR@$r$ and NDCG@$r$ for $r \in \{5, 10, 20\}$) computed over the entire item set without negative sampling [20]. Detailed experimental setups, including dataset statistics and optimal hyperparameter configurations, are provided in Appendix D. The source code is publicly available at `https://github.com/yehjin-shin/TV-Rec`.

Table 2: Performance comparison of different methods. Best results are in **bold** and second-best results are underlined. 'Improv.' indicates the relative improvement against the best baseline.

| Datasets | Metric | Caser | GRU4Rec | SASRec | BERT4Rec | NextItNet | FMLPRec | DuoRec | LRURec | AdaMCT | BSARec | TV-Rec | Improv. |
|---|---|---|---|---|---|---|---|---|---|---|---|---|---|
| Beauty | HR@5 | 0.0149 | 0.0170 | 0.0368 | 0.0491 | 0.0549 | 0.0423 | 0.0680 | 0.0648 | 0.0675 | 0.0714 | **0.0721** | 0.98% |
| | HR@10 | 0.0253 | 0.0307 | 0.0574 | 0.0742 | 0.0779 | 0.0639 | 0.0944 | 0.0889 | 0.0925 | 0.0990 | **0.1017** | 2.73% |
| | HR@20 | 0.0416 | 0.0499 | 0.0860 | 0.1079 | 0.1100 | 0.0949 | 0.1279 | 0.1197 | 0.1299 | 0.1393 | **0.1403** | 0.72% |
| | NDCG@5 | 0.0089 | 0.0105 | 0.0241 | 0.0318 | 0.0392 | 0.0272 | 0.0485 | 0.0472 | 0.0489 | 0.0501 | **0.0513** | 2.40% |
| | NDCG@10 | 0.0122 | 0.0149 | 0.0307 | 0.0399 | 0.0467 | 0.0341 | 0.0570 | 0.0549 | 0.0569 | 0.0590 | **0.0608** | 3.05% |
| | NDCG@20 | 0.0164 | 0.0198 | 0.0379 | 0.0484 | 0.0547 | 0.0419 | 0.0654 | 0.0627 | 0.0664 | 0.0691 | **0.0705** | 2.03% |
| Sports | HR@5 | 0.0091 | 0.0131 | 0.0215 | 0.0279 | 0.0311 | 0.0222 | 0.0390 | 0.0351 | 0.0386 | 0.0422 | **0.0431** | 2.13% |
| | HR@10 | 0.0147 | 0.0211 | 0.0319 | 0.0434 | 0.0458 | 0.0358 | 0.0549 | 0.0502 | 0.0544 | 0.0623 | **0.0635** | 1.93% |
| | HR@20 | 0.0253 | 0.0347 | 0.0485 | 0.0658 | 0.0682 | 0.0549 | 0.0779 | 0.0698 | 0.0769 | 0.0865 | **0.0880** | 1.73% |
| | NDCG@5 | 0.0064 | 0.0084 | 0.0142 | 0.0182 | 0.0212 | 0.0148 | 0.0276 | 0.0242 | 0.0272 | 0.0296 | **0.0298** | 0.68% |
| | NDCG@10 | 0.0082 | 0.0110 | 0.0175 | 0.0232 | 0.0260 | 0.0191 | 0.0328 | 0.0291 | 0.0322 | 0.0361 | **0.0363** | 0.55% |
| | NDCG@20 | 0.0109 | 0.0144 | 0.0217 | 0.0288 | 0.0316 | 0.0239 | 0.0385 | 0.0340 | 0.0379 | 0.0422 | **0.0425** | 0.71% |
| Yelp | HR@5 | 0.0131 | 0.0137 | 0.0165 | 0.0243 | 0.0247 | 0.0195 | 0.0277 | 0.0240 | 0.0239 | 0.0260 | **0.0290** | 4.69% |
| | HR@10 | 0.0230 | 0.0240 | 0.0267 | 0.0411 | 0.0423 | 0.0313 | 0.0450 | 0.0396 | 0.0404 | 0.0446 | **0.0474** | 5.33% |
| | HR@20 | 0.0388 | 0.0412 | 0.0445 | 0.0681 | 0.0694 | 0.0518 | 0.0730 | 0.0656 | 0.0670 | 0.0718 | **0.0777** | 6.44% |
| | NDCG@5 | 0.0080 | 0.0086 | 0.0103 | 0.0154 | 0.0151 | 0.0122 | 0.0179 | 0.0151 | 0.0153 | 0.0162 | **0.0186** | 3.91% |
| | NDCG@10 | 0.0112 | 0.0119 | 0.0135 | 0.0208 | 0.0208 | 0.0160 | 0.0234 | 0.0201 | 0.0206 | 0.0222 | **0.0245** | 4.70% |
| | NDCG@20 | 0.0151 | 0.0162 | 0.0180 | 0.0275 | 0.0276 | 0.0211 | 0.0304 | 0.0266 | 0.0272 | 0.0290 | **0.0321** | 5.59% |
| LastFM | HR@5 | 0.0303 | 0.0339 | 0.0422 | 0.0358 | 0.0431 | 0.0450 | 0.0404 | 0.0358 | 0.0468 | 0.0505 | **0.0596** | 18.02% |
| | HR@10 | 0.0459 | 0.0394 | 0.0670 | 0.0606 | 0.0624 | 0.0670 | 0.0587 | 0.0532 | 0.0716 | 0.0716 | **0.0853** | 19.13% |
| | HR@20 | 0.0606 | 0.0550 | 0.0972 | 0.0908 | 0.0936 | 0.1000 | 0.0872 | 0.0807 | 0.1018 | 0.1119 | **0.1202** | 7.42% |
| | NDCG@5 | 0.0222 | 0.0231 | 0.0301 | 0.0213 | 0.0264 | 0.0321 | 0.0276 | 0.0257 | 0.0330 | 0.0348 | **0.0402** | 15.52% |
| | NDCG@10 | 0.0269 | 0.0249 | 0.0382 | 0.0291 | 0.0325 | 0.0392 | 0.0336 | 0.0312 | 0.0409 | 0.0405 | **0.0484** | 18.34% |
| | NDCG@20 | 0.0306 | 0.0288 | 0.0458 | 0.0366 | 0.0402 | 0.0475 | 0.0407 | 0.0380 | 0.0485 | 0.0514 | **0.0572** | 11.28% |
| ML-1M | HR@5 | 0.1033 | 0.1225 | 0.1406 | 0.1651 | 0.1858 | 0.1329 | 0.1821 | 0.1916 | 0.1773 | 0.1909 | **0.2013** | 5.06% |
| | HR@10 | 0.1671 | 0.1925 | 0.2199 | 0.2442 | 0.2724 | 0.2089 | 0.2690 | 0.2848 | 0.2560 | 0.2798 | **0.2904** | 1.97% |
| | HR@20 | 0.2598 | 0.2906 | 0.3250 | 0.3459 | 0.3853 | 0.3212 | 0.3757 | 0.3886 | 0.3647 | 0.3844 | **0.4079** | 4.97% |
| | NDCG@5 | 0.0663 | 0.0779 | 0.0920 | 0.1077 | 0.1264 | 0.0861 | 0.1226 | 0.1339 | 0.1185 | 0.1286 | **0.1371** | 2.39% |
| | NDCG@10 | 0.0868 | 0.1006 | 0.1174 | 0.1332 | 0.1543 | 0.1105 | 0.1507 | 0.1640 | 0.1438 | 0.1573 | **0.1658** | 1.10% |
| | NDCG@20 | 0.1101 | 0.1253 | 0.1438 | 0.1588 | 0.1829 | 0.1388 | 0.1776 | 0.1901 | 0.1711 | 0.1836 | **0.1955** | 2.84% |
| Foursquare | HR@5 | 0.0139 | 0.0148 | 0.0139 | 0.0139 | 0.0129 | 0.0120 | 0.0139 | 0.0148 | 0.0157 | 0.0148 | **0.0175** | 11.46% |
| | HR@10 | 0.0175 | 0.0157 | 0.0185 | 0.0157 | 0.0175 | 0.0175 | 0.0185 | 0.0166 | 0.0185 | 0.0212 | **0.0259** | 22.17% |
| | HR@20 | 0.0268 | 0.0231 | 0.0268 | 0.0231 | 0.0203 | 0.0240 | 0.0240 | 0.0212 | 0.0259 | 0.0277 | **0.0314** | 13.36% |
| | NDCG@5 | 0.0099 | 0.0110 | 0.0102 | 0.0108 | 0.0093 | 0.0076 | 0.0110 | 0.0110 | 0.0105 | 0.0111 | **0.0134** | 20.72% |
| | NDCG@10 | 0.0110 | 0.0113 | 0.0117 | 0.0113 | 0.0108 | 0.0094 | 0.0124 | 0.0116 | 0.0113 | 0.0130 | **0.0161** | 23.85% |
| | NDCG@20 | 0.0133 | 0.0132 | 0.0137 | 0.0132 | 0.0115 | 0.0110 | 0.0139 | 0.0127 | 0.0131 | 0.0147 | **0.0176** | 19.73% |

Table 3: Results of long-range modeling performance.

| Datasets | Metric | Caser | GRU4Rec | SASRec | BERT4Rec | NextItNet | FMLPRec | DuoRec | LRURec | AdaMCT | BSARec | TV-Rec | Improv. |
|---|---|---|---|---|---|---|---|---|---|---|---|---|---|
| ML-1M | HR@5 | 0.1109 | 0.1518 | 0.1558 | 0.1730 | 0.1978 | 0.1397 | 0.1930 | 0.2233 | 0.1760 | 0.1949 | **0.2255** | 0.99% |
| | HR@10 | 0.1869 | 0.2374 | 0.2399 | 0.2573 | 0.2882 | 0.2296 | 0.2795 | 0.3175 | 0.2619 | 0.2917 | **0.3232** | 1.80% |
| | HR@20 | 0.2942 | 0.3455 | 0.3551 | 0.3695 | 0.3970 | 0.3462 | 0.3854 | 0.4205 | 0.3695 | 0.4005 | **0.4306** | 2.40% |
| | NDCG@5 | 0.0696 | 0.0981 | 0.1014 | 0.1147 | 0.1334 | 0.0885 | 0.1292 | 0.1516 | 0.1167 | 0.1327 | **0.1572** | 3.69% |
| | NDCG@10 | 0.0939 | 0.1256 | 0.1285 | 0.1418 | 0.1627 | 0.1175 | 0.1571 | 0.1820 | 0.1443 | 0.1639 | **0.1886** | 3.63% |
| | NDCG@20 | 0.1209 | 0.1528 | 0.1576 | 0.1701 | 0.1901 | 0.1468 | 0.1838 | 0.2079 | 0.1715 | 0.1913 | **0.2157** | 3.75% |
| Foursquare | HR@5 | 0.0139 | 0.0120 | 0.0111 | 0.0102 | 0.0083 | 0.0120 | 0.0120 | 0.0129 | 0.0120 | 0.0129 | **0.0148** | 6.47% |
| | HR@10 | 0.0194 | 0.0157 | 0.0175 | 0.0157 | 0.0166 | 0.0148 | 0.0194 | 0.0139 | 0.0157 | 0.0175 | **0.0212** | 9.28% |
| | HR@20 | 0.0231 | 0.0194 | 0.0295 | 0.0240 | 0.0259 | 0.0194 | 0.0286 | 0.0185 | 0.0305 | 0.0305 | **0.0323** | 5.90% |
| | NDCG@5 | 0.0105 | 0.0099 | 0.0085 | 0.0078 | 0.0068 | 0.0087 | 0.0078 | 0.0099 | 0.0094 | 0.0089 | **0.0108** | 2.86% |
| | NDCG@10 | 0.0123 | 0.0111 | 0.0106 | 0.0096 | 0.0095 | 0.0096 | 0.0102 | 0.0102 | 0.0106 | 0.0103 | **0.0129** | 4.88% |
| | NDCG@20 | 0.0133 | 0.0120 | 0.0136 | 0.0117 | 0.0118 | 0.0108 | 0.0126 | 0.0114 | 0.0142 | 0.0135 | **0.0158** | 11.27% |

## 4.2 Overall Performance

**Sequential Recommendation Results.** As shown in Table 2, TV-Rec outperforms all baseline methods, with an average accuracy improvement of 7.49% over the strongest baselines. The improvements are particularly significant on LastFM (19.13% on HR@10 and 18.34% on NDCG@10) and Foursquare (22.17% on HR@10 and 23.85% on NDCG@10). On larger datasets like ML-1M, TV-Rec still shows gains with improvements of 5.06% on HR@5 and 2.39% on NDCG@5. However, for E-commerce datasets like Beauty and Sports, the improvements are more subtle but still consistent (0.98% and 2.13% on HR@5, respectively). Among the baselines, recent hybrid methods that combine convolution and self-attention, such as AdaMCT and BSARec, show strong performance. BSARec achieves the second-best results in many cases. For ML-1M, LRURec shows competitive results with its specialized architecture for long sequences, while DuoRec achieves second-best performance for Yelp. Despite these strong baselines, TV-Rec consistently outperforms them by significant margins.

Table 4: Results of performance comparison with GNN-based methods.

| Methods | Beauty | | Sports | | Yelp | | LastFM | | ML-1M | | Foursquare | |
|---|---|---|---|---|---|---|---|---|---|---|---|---|
| | H@20 | N@20 | H@20 | N@20 | H@20 | N@20 | H@20 | N@20 | H@20 | N@20 | H@20 | N@20 |
| **TV-Rec** | **0.1403** | **0.0705** | **0.0880** | **0.0425** | **0.0777** | **0.0321** | **0.1202** | **0.0572** | **0.4079** | **0.1955** | **0.0314** | **0.0176** |
| SR-GNN | 0.0847 | 0.0374 | 0.0517 | 0.0224 | 0.0609 | 0.0252 | 0.0872 | 0.0379 | 0.2940 | 0.1390 | 0.0222 | 0.0137 |
| GC-SAN | 0.1059 | 0.0546 | 0.0608 | 0.0289 | 0.0635 | 0.0260 | 0.0807 | 0.0394 | 0.3255 | 0.1611 | 0.0212 | 0.0131 |
| GCL4SR | 0.1206 | 0.0601 | 0.0744 | 0.0356 | 0.0684 | 0.0276 | 0.0908 | 0.0398 | 0.3381 | 0.1607 | 0.0185 | 0.0123 |

Table 5: Ablation studies.

| Methods | Beauty | | Sports | | Yelp | | LastFM | | ML-1M | | Foursquare | |
|---|---|---|---|---|---|---|---|---|---|---|---|---|
| | H@20 | N@20 | H@20 | N@20 | H@20 | N@20 | H@20 | N@20 | H@20 | N@20 | H@20 | N@20 |
| **TV-Rec** | 0.1403 | **0.0705** | **0.0880** | **0.0425** | **0.0777** | **0.0321** | **0.1202** | **0.0572** | **0.4079** | **0.1955** | **0.0314** | **0.0176** |
| (1) Positional Embedding | **0.1408** | 0.0702 | 0.0842 | 0.0396 | 0.0763 | 0.0320 | 0.1018 | 0.0496 | 0.4017 | 0.1936 | 0.0313 | 0.0160 |
| (2) Basic Graph Filter | 0.1402 | 0.0692 | 0.0857 | 0.0408 | 0.0747 | 0.0307 | 0.1165 | 0.0543 | 0.3974 | 0.1933 | 0.0212 | 0.0113 |
| (3) Identity Basis | 0.1400 | 0.0698 | 0.0851 | 0.0410 | 0.0765 | 0.0317 | 0.1138 | 0.0539 | 0.4015 | 0.1930 | 0.0277 | 0.0141 |
| (4) Basis Normalization | 0.1336 | 0.0689 | 0.0841 | 0.0412 | 0.0634 | 0.0264 | 0.0963 | 0.0418 | 0.3985 | 0.1912 | 0.0305 | 0.0144 |

**Long-Range Sequential Recommendation Results.** To examine the performance of TV-Rec on long-range dependencies, we additionally conduct experiments on ML-1M and Foursquare, which have a long average interaction length, by setting the maximum length $N$ to 200. As shown in Table 3, TV-Rec outperforms all baseline models in long-range SR tasks. Our model achieves an average improvement of 4.74% on all metrics compared to the top baseline performances. The improvement is particularly significant in NDCG metrics, with TV-Rec showing up to 11.27% gain in NDCG@20 for Foursquare. For ML-1M, we observe that LRURec performs strongest among baselines due to its linear recurrent unit designed for long sequences. For Foursquare, Caser and AdaMCT show the highest baseline performance. The superior results of TV-Rec on these settings show the effectiveness of our time-variant filters on extended user interaction histories. This shows the ability of our approach to maintain recommendation accuracy even when processing sequences with hundreds of interactions.

**Comparison to GNN-based Methods.** To further examine its effectiveness, we also compare TV-Rec with three representative GNN-based sequential recommendation models: SR-GNN [35], GC-SAN [37], and GCL4SR [45], under the same experimental settings described in Section 4.1. While most GNN-based recommendation methods are tailored for collaborative filtering on static user–item graphs, these GNN-based sequential models utilize item-transition graphs to model local or global item dependencies. As shown in Table 4, TV-Rec consistently outperforms all GNN-based sequential models, confirming the advantage of its time-variant filtering design. By operating directly on sequence positions instead of propagating messages over item-transition graphs, TV-Rec achieves a more efficient representation of temporal dependencies.

## 4.3 Ablation Studies

We conduct ablation studies to validate the design choices of TV-Rec. The results are shown in Table 5. (1) First, we test an ablation model with added learnable positional embeddings. Unlike recent SR models [29, 6, 17], TV-Rec performs effectively without positional embeddings because our time-variant filter inherently captures position-specific information. The first ablation model yields inconsistent results on all datasets. (2) Second, we define the second ablation model by replacing our time-variant filter with a basic graph filter. This ablation model degrades the performance, which confirms the effectiveness of our method as discussed in Sec. 4.5. (3) Third, to validate our filter construction method (Eq. 8), we define the third ablation model by setting $\mathbf{B}$ as an identity matrix, which makes $\mathbf{H}$ equal to $\mathbf{C}$. This ablation model degrades results compared to our filter construction method. (4) Fourth, we compare against the fourth ablation model that uses the basis matrix $\mathbf{B}$ directly without normalization, demonstrating the effectiveness of our normalization approach. These results prove that each component of our design contributes to the superiority of TV-Rec.

## 4.4 Parameter Sensitivity

We analyze the sensitivity of the parameters $m$ and dropout rate $p$, as shown in Figs. 3 and 4. All other hyperparameters are fixed at their optimal values. Additional results, including sensitivity to filter order $K$ and weight decay $\alpha$, are provided in Appendix E.

**Sensitivity to $m$.** We employ a basis matrix **B** in the time-variant graph filter (Eq. (8)), where $m$ controls the number of basis vectors, influencing both expressiveness and computational cost. As shown in Fig. 3, performance on Beauty peaks at $m = 32$ but drops sharply for smaller values, indicating the need for richer representations. Conversely, LastFM achieves its best results at $m = 8$, with performance degrading as $m$ increases, suggesting that a compact representation suffices for this dataset.

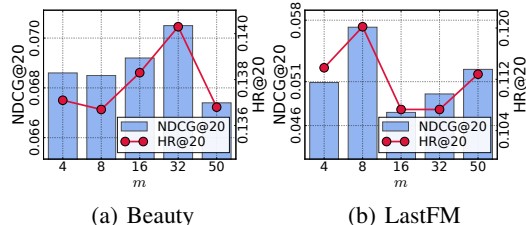

(a) Beauty          (b) LastFM

Figure 3: Sensitivity to the number of basis vectors $m$.

**Sensitivity to $p$.** The effect of dropout rate $p$ on Sports and Foursquare is shown in Fig. 4. For Sports, larger $p$ values improve performance, while for Foursquare, smaller $p$ values work better. Datasets with fewer interactions tend to benefit from higher dropout rates, which help prevent overfitting by encouraging more general representations.

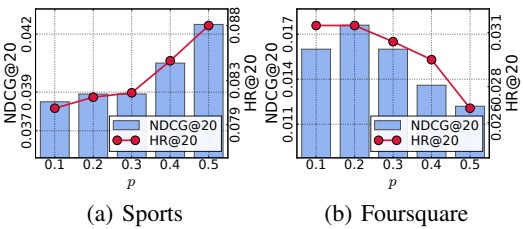

(a) Sports          (b) Foursquare

Figure 4: Sensitivity to dropout rate $p$.

## 4.5 Analyzing Filter Behavior and Case Study

To understand why our time-variant filter works better than fixed basic graph filter (Eq.(3)), we analyze their learned representations via visualizations and a case study. Fig. 5 reveals the key difference between approaches: the basic graph filter applies the same filter to all nodes, while our time-variant filter applies different filters to each node.

Both filters assign higher weights to lower shifts, meaning they give stronger weights to recent items. However, unlike the basic graph filter, the time-variant convolutional filter starts with similar weights for the early-stage nodes, reflecting the model's focus on understanding overall patterns. As the sequence progresses, the filter's weights shift to emphasize recent items, allowing the filter to capture temporal shifts more accurately. This shows the time-variant filter's ability to adapt to both early and recent changes, boosting performance.

Our case study on ML-1M (see Fig. 6) shows this advantage in practice. The ba-

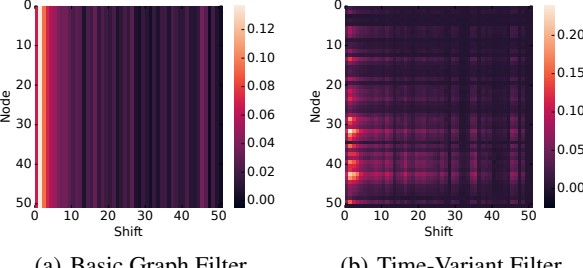

(a) Basic Graph Filter          (b) Time-Variant Filter

Figure 5: Visualization of learned graph filters on LastFM. The x-axis denotes the number of shifts in graph convolution, while the y-axis represents individual nodes, with higher numbers indicating more recent time points.

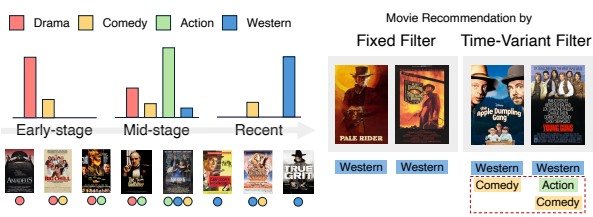

Figure 6: Case Study on ML-1M.

sic graph filter focuses solely on *Western* films from recent interactions, while our time-variant filter captures both user's recent *Western* interest and their broader *Comedy* preference. These findings confirm our statement that applying different filtering operations at different temporal positions is important for effective sequential recommendation.

### 4.6 Model Complexity and Runtime Analyses

To evaluate the efficiency of TV-Rec, we analyze the number of parameters and inference time. The results across the full dataset are provided in the Appendix G. As shown in Fig. 7, TV-Rec achieves the best balance between performance and computational efficiency. Among top-performing methods, TV-Rec has the fastest inference time with the smallest number of parameters, compared to hybrid models that combine convolution and self-attention, such as AdaMCT and BSARec. Compared to SASRec and BERT4Rec, which use only self-attention, TV-Rec provides faster inference time and superior recommendation accuracy. While FMLPRec runs slightly faster with its simple architecture, it achieves this through a basic graph filter that degrades recommendation quality. Considering the improved performance of TV-Rec, the marginally higher inference time than FMLPRec is acceptable.

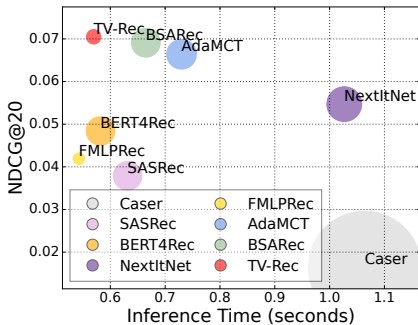

Figure 7: Comparison of model inference time and NDCG@20 on Beauty. The size of each circle corresponds to the number of parameters.

## 5 Related Work

### 5.1 Sequential Recommendation

SR has evolved from early approaches that used Markov chains [25] and RNNs [14] to model sequential dependencies. Transformer-based models such as SASRec [18] and BERT4Rec [30] use self-attention to capture global dependencies and establish new performance benchmarks. Recent advanced methods have emerged to address specific challenges and efficiency-performance trade-offs in SR. FMLPRec [47] proposes a filter-enhanced MLP to eliminate frequency domain noise, while FEARec [6] and DuoRec [24] use contrastive learning approaches for better sequence representation. AC-TSR [48] calibrates unreliable attention weights generated by Transformer-based models. LRURec [43] explores linear recurrent units to balance efficiency and performance. In addition, SR models [22, 44, 40, 33] based on state space models [11, 10, 9] have been explored for potential in SR.

### 5.2 Hybrid Approaches and Convolution in SR

Recent research has shown that convolution-based methods can serve as competitive alternatives to existing SR methods [31, 38, 39, 47, 17, 29, 5]. The first to use convolution in SR was Caser [31], which treats user-item interactions as images for 2D convolutions, followed by NextItNet [42], which uses dilated 1D convolutions. FMLPRec [47] incorporated Fourier transforms within an all-MLP architecture to enhance sequence representations. AdaMCT [17] incorporates 1D convolution into Transformer-based recommendation model to capture both long-term and short-term user preferences. BSARec [29] has recently achieved state-of-the-art results by addressing the limitations of self-attention through the application of convolution using the Fourier transform. However, these models typically require self-attention to achieve optimal performance. Our work differs by introducing time-variant convolutional filters that achieve high performance without relying on self-attention.

## 6 Conclusion

We focus on the inherent limitations of conventional convolutional filters. Since these filters are fixed, they may struggle to capture the complex patterns required for SR. To address this issue, we introduce TV-Rec, which uses time-variant convolutional filters that apply different filters to each data point. TV-Rec achieves high performance without relying on self-attention, even in long-range modeling. Our method also benefits from fast inference times due to its linear operator nature. We validated the effectiveness and efficiency of TV-Rec through extensive experiments on 6 datasets. In future work, we plan to explore both the theoretical and practical relationships between our time-variant filter and recent advances in state space models, which have demonstrated strong connections with convolutional filters. We leave limitations in Appendix J.

## Acknowledgments

This work was partly supported by the Institute for Information & Communications Technology Planning & Evaluation (IITP) grants funded by the Korean government (MSIT) (No. RS-2022-II220113, Developing a Sustainable Collaborative Multi-modal Lifelong Learning Framework), Samsung Electronics Co., Ltd. (No. G01240136, KAIST Semiconductor Research Fund (2nd)), and Samsung Research Funding & Incubation Center of Samsung Electronics under Project Number SRFC-IT2402-08.

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

# Supplementary Materials for "TV-Rec: Time-Variant Convolutional Filter for Sequential Recommendation"

## A Proof of Frequency Response of Node-Variant Graph Filters

In this section, we provide a detailed proof of the frequency response of the node-variant graph filter $\mathbf{G}_{nv}$ as stated in Eq. (5), following the derivation in [7].

*Proof.* We begin by considering the definition of the node-variant graph filter. Let $\mathbf{x} \in \mathbb{R}^N$ represent the input graph signal, $\mathbf{S} \in \mathbb{R}^{N \times N}$ represent the shift operator, and let $\mathbf{h}_k$ be a vector of filter taps. By using the node-variant graph filter $\mathbf{G}_{nv}$, the output graph signal $\mathbf{y}$ is calculated as follows:

$$\mathbf{y} = \mathbf{G}_{nv}\mathbf{x} = \Big( \sum_{k=0}^{K} \texttt{diag}(\mathbf{h}_k)\mathbf{S}^k \Big)\mathbf{x}, \tag{14}$$

where $\texttt{diag}(\cdot)$ indicates constructing a diagonal matrix from a vector, and $K$ is the order of the filter. Recall that the frequency response of the input signal $\tilde{\mathbf{x}}$ is $\mathbf{U}^\top\mathbf{x}$, and similarly $\tilde{\mathbf{y}} = \mathbf{U}^\top\mathbf{y}$, where the shift operator $\mathbf{S}$ is eigen-decomposed by $\mathbf{S} = \mathbf{U}\,\texttt{diag}(\boldsymbol{\lambda})\mathbf{U}^\top$, which results in:

$$\tilde{\mathbf{y}} = \mathbf{U}^\top\mathbf{y} = \mathbf{U}^\top\Big( \sum_{k=0}^{K} \texttt{diag}(\mathbf{h}_k)\mathbf{S}^k \Big)\mathbf{x} \tag{15}$$

$$= \mathbf{U}^\top\sum_{k=0}^{K} \texttt{diag}(\mathbf{h}_k)\mathbf{U}\,\texttt{diag}(\boldsymbol{\lambda}^k)\tilde{\mathbf{x}}, \tag{16}$$

where $\boldsymbol{\lambda}^k \in \mathbb{C}^N$ is the $k$-th column of a Vandermonde matrix $\boldsymbol{\Lambda} \in \mathbb{C}^{N \times (K+1)}$ given by $\boldsymbol{\Lambda}_{ik} = [\boldsymbol{\lambda}^k]_i = \lambda_i^k$. The set of filter taps $\mathbf{h}_k$ can be represented in matrix form as $\mathbf{H} \in \mathbb{C}^{N \times (K+1)}$. Each element of the summation in Eq. (16) is as follows:

$$\left[ \sum_{k=0}^{K} \texttt{diag}(\mathbf{h}_k)\mathbf{U}\,\texttt{diag}(\boldsymbol{\lambda}^k) \right]_{ij} = \sum_{k=0}^{K} h_k^{(i)}\lambda_j^k u_{ij} \tag{17}$$

$$= u_{ij}\sum_{k=0}^{K} h_k^{(i)}\lambda_j^k, \tag{18}$$

where $h_k^{(i)}$ is the $i$-th row and $k$-th column of $\mathbf{H}$. Using the matrices $\mathbf{H}$ and $\boldsymbol{\Lambda}$, it holds that $\mathbf{H}\boldsymbol{\Lambda}^\top \in \mathbb{C}^{N \times N}$:

$$[\mathbf{H}\boldsymbol{\Lambda}^\top]_{ij} = \sum_{k=0}^{K} h_k^{(i)}\lambda_j^k. \tag{19}$$

Substituting Eq. (19) into Eq. (18) results in:

$$\left[ \sum_{k=0}^{K} \texttt{diag}(\mathbf{h}_k)\mathbf{U}\,\texttt{diag}(\boldsymbol{\lambda}^k) \right]_{ij} = [\mathbf{U} \circ \mathbf{H}\boldsymbol{\Lambda}^\top]_{ij}, \tag{20}$$

where $\circ$ denotes element-wise multiplication between matrices.

Therefore, Eq. (16) becomes:

$$\tilde{\mathbf{y}} = \tilde{\mathbf{G}}_{nv}\tilde{\mathbf{x}} = \mathbf{U}^\top(\mathbf{U} \circ (\mathbf{H}\boldsymbol{\Lambda}^\top))\tilde{\mathbf{x}}. \tag{21}$$

This completes the proof.

$\square$

# B Theoretical Justification for DCG-Based Filtering

This section theoretically shows that spectral filtering on the zero-padded DCG is equivalent to filtering on the line graph, ensuring no backward information flow.

To perform spectral filtering, the model must apply GFT, which requires eigen decomposition of the graph shift operator (i.e., the adjacency or Laplacian matrix). However, when the sequence is modeled as a line graph, the resulting adjacency matrix has rank exactly $N - 1$ because the first node (i.e., the most past item) has no incoming edges. This results in one row of the matrix being entirely zero, making it defective and thus non-diagonalizable.

To resolve this, we model the sequence as a DCG, which differs from the line graph by a single edge connecting the last node to the first. This addition makes the adjacency matrix circulant, ensuring diagonalizability and enabling spectral filtering in the Fourier domain. However, the added edge introduces a backward connection from the future to the past, which could lead to information leakage in the reverse temporal direction.

To prevent reverse information flow while maintaining spectral tractability, we adopt a padding strategy. Specifically, we pad the sequence $\mathbf{x} \in \mathbb{R}^N$ with $K$ zeros by forming the extended vector:

$$\tilde{\mathbf{x}} = \begin{bmatrix} \mathbf{x} \\ \mathbf{0} \end{bmatrix} \in \mathbb{R}^{N+K}, \tag{22}$$

where $\mathbf{0} \in \mathbb{R}^K$ is the zero vector. Using the circulant shift operator $\mathbf{S} \in \mathbb{R}^{(N+K) \times (N+K)}$ that represents the DCG, we define a spectral filter of order $K$ as follows:

$$g(\mathbf{S}) = \sum_{k=0}^{K} h_k \mathbf{S}^k, \tag{23}$$

where $h_k$ denotes the filter coefficients. The filtered output $\tilde{\mathbf{y}}$ is then computed by applying the filter to the padded input:

$$\tilde{\mathbf{y}} = g(\mathbf{S})\tilde{\mathbf{x}}. \tag{24}$$

We obtain a result identical to applying the same filter on the original sequence $\mathbf{x}$ modeled as a line graph by extracting the first $N$ elements of $\tilde{\mathbf{y}}$ as follows:

$$\mathbf{y} = [\tilde{\mathbf{y}}]_{1:N}, \tag{25}$$

where $1 : N$ denotes taking the first $N$ elements of the vector. This equivalence holds because the last $K$ entries of $\tilde{\mathbf{x}}$ are zeros. Although the circulant matrix $\mathbf{S}$ performs circular shifts, the filter involves powers of $\mathbf{S}$ up to order $K$, so these zero entries do not influence the first $N$ output values. Consequently, this effectively blocks any cyclic information flow that would lead to backward leakage.

As a result, spectral filtering on the zero-padded DCG provides the correct filtering output equivalent to causal convolution on a line graph, while benefiting from the computational efficiency and diagonalizability of circulant matrices.

# C Equivalence of Positional Encoding and Graph Fourier Basis

Unlike Transformers, which require explicit positional encoding (e.g., sinusoidal or learnable), TV-Rec captures positional information inherently through spectral decomposition on DCG. This section provides a formal connection between positional encodings in Transformer and the graph Fourier basis used in TV-Rec.

**Sinusoidal Positional Encoding in Transformer.** The original Transformer model uses sinusoidal functions to encode absolute position $pos$ as follows:

$$\mathrm{PE}_{(pos,2i)} = \sin\left(\frac{pos}{10000^{2i/d}}\right), \quad \mathrm{PE}_{(pos,2i+1)} = \cos\left(\frac{pos}{10000^{2i/d}}\right),$$

where $d$ denotes the embedding dimension. This produces a set of periodic signals with varying frequencies that form a basis for encoding positional variation.

**Graph Fourier Basis in TV-Rec.** In TV-Rec, we define the shift operator $\mathbf{S}$ as the adjacency matrix of a directed cyclic graph. Its eigen-decomposition yields the graph Fourier basis $\mathbf{U} \in \mathbb{C}^{N \times N}$, where:

$$\mathbf{U}_{kn} = \frac{1}{\sqrt{N}} e^{-i2\pi kn/N} = \frac{1}{\sqrt{N}} \left[ \cos\left(\frac{2\pi kn}{N}\right) - i\sin\left(\frac{2\pi kn}{N}\right) \right].$$

This corresponds to the DFT basis, which is an orthonormal set of complex exponential spanning $\mathbb{R}^N$ (or $\mathbb{C}^N$).

**Equivalence in Representational Capacity.** While the frequency components used in Transformer positional encodings are sampled on a logarithmic scale and those in GFT are linearly spaced, both sets of basis functions are composed of trigonometric functions. As such, they span the same space of length-$N$ periodic signals. Formally, the real-valued DFT basis used in TV-Rec corresponds to $\sin\left(\frac{2\pi kn}{N}\right)$ and $\cos\left(\frac{2\pi kn}{N}\right)$ for various $k$, which can represent any finite-length sinusoidal signal, including those used in Transformer encodings. Therefore, although the sampling strategies differ, the span of both bases covers the same function space, which indicates that the spectral basis used in TV-Rec is functionally equivalent to the sinusoidal encodings in Transformer models, in the sense that both span the same space of position-dependent trigonometric functions.

**Implication for Positional Encoding.** TV-Rec projects the input sequence $x$ to the GFT domain as $\tilde{x} = \mathbf{U}^{\top} x$, and reconstructs it via $x = \mathbf{U}\tilde{x}$, thereby implicitly encoding frequency-based positional variations. The time-variant filter then modulates these frequency components per position, which makes explicit positional embeddings unnecessary, as position-sensitive modulation is already embedded in the spectral filtering process. TV-Rec's design leverages the spectral basis to achieve the same functional role as positional encoding in Transformers. It achieves this without requiring explicit embeddings, while retaining full expressiveness in modeling temporal variation.

# D Detailed Experimental Settings

## D.1 Datasets

We evaluate our model using 6 SR datasets that vary in sparsity and domain. We follow the data pre-processing procedures outlined in [46, 47], considering all reviews and ratings as implicit feedback. Detailed statistics can be found in Table 6.

- **Amazon Beauty** and **Sports** are Amazon datasets of product reviews from [23], widely used for SR. For this study, we use the "*Beauty*" and "*Sports and Outdoors*" categories.
- **Yelp**[4] is a popular business recommendation dataset. We use records from after 2019/01/01 due to its large size.
- **LastFM**[5] includes artist listening records and is used to recommend musicians to users.
- **ML-1M** [12] is a movie recommendation dataset from MovieLens[6]. It is commonly used to evaluate recommendation algorithms due to its detailed user interaction data.
- **Foursquare**[7] provides user check-ins across New York city over 10 months (April 2012 to February 2013).

## D.2 Baselines

To evaluate the performance of our method, we compare it with the following ten SR baseline methods:

- **Caser** [31] is a CNN-based model that captures complex user patterns through horizontal and vertical convolutions.

---

[4]https://www.yelp.com/dataset
[5]https://grouplens.org/datasets/hetrec-2011/
[6]https://grouplens.org/datasets/movielens/
[7]https://sites.google.com/site/yangdingqi/home/foursquare-dataset

Table 6: Statistics of the processed datasets.

| | # Users | # Items | # Interactions | Avg. Length | Sparsity |
|---|---|---|---|---|---|
| Beauty | 22,363 | 12,101 | 198,502 | 8.9 | 99.93% |
| Sports | 25,598 | 18,357 | 296,337 | 8.3 | 99.95% |
| Yelp | 30,431 | 20,033 | 316,354 | 10.4 | 99.95% |
| LastFM | 1,090 | 3,646 | 52,551 | 48.2 | 98.68% |
| ML-1M | 6,041 | 3,417 | 999,611 | 165.5 | 95.16% |
| Foursquare | 1,083 | 9,989 | 179,468 | 165.7 | 98.34% |

- **GRU4Rec** [14] is a GRU-based model that captures temporal dynamics and patterns in user interactions.
- **SASRec** [18] is a Transformer-based model that uses a multi-head self-attention mechanism.
- **BERT4Rec** [30] is a bidirectional Transformer-based model, using a masked item training scheme.
- **NextItNet** [42] is a CNN-based model that uses dilated convolutions and residual connections to capture both short- and long-range dependencies in user behavior sequences.
- **FMLPRec** [47] uses Fourier Transform and learnable filters in an all-MLP architecture to reduce noise and enhance sequence representations.
- **DuoRec** [24] employs model-level augmentation and sementic positive samples for contrastive learning, using SASRec as its base model.
- **LRURec**[43] uses linear recurrent units for rapid inference and recursive parallelization.
- **AdaMCT** [17] is a hybrid model combining Transformer attention with local convolutional filters to capture long- and short-term user preferences.
- **BSARec** [29] is a hybrid model that combines Transformer self-attention with the Fourier Transform to address the oversmoothing problem.
- **SR-GNN** [35] is a session-based recommendation model which converts user sessions into graphs and applies GNNs to capture item transition relationships.
- **GC-SAN** [37] is a GNN-based model that dynamically builds a graph for each sequence and combines GNN with self-attention to model both local and long-range item dependencies.
- **GCL4SR** [45] is a GNN-based model that constructs a global item transition graph across all users and uses graph contrastive learning to integrate global and local context.

### D.3 Metrics

To evaluate the recommendation performance, we use widely adopted Top-$r$ metrics, HR@$r$ (Hit Rate) and NDCG@$r$ (Normalized Discounted Cumulative Gain), with $r$ set to 5, 10, and 20. For a fair comparison, we examine the ranking results across the entire item set without negative sampling [20].

### D.4 Implementation Details

All experiments, including the baselines, were conducted using the following software and hardware configurations: UBUNTU 20.04.6 LTS, PYTHON 3.9.7, PYTORCH 2.2.2, CUDA 11.1.74, and NVIDIA Driver 550.54.14. The hardware setup included dual INTEL XEON CPUs and an NVIDIA RTX A6000 GPU.

**Hyperparameters for Standard Sequential Recommendation.** We determine the optimal hyperparameters for the baselines according to their suggested settings. The experiments are performed with the following hyperparameters: learning rates of $\{5 \times 10^{-4}, 1 \times 10^{-3}\}$, orthogonal regularization coefficient $\alpha$ of $\{0, 1 \times 10^{-3}, 1 \times 10^{-5}\}$, dropout rates $p$ of $\{0.1, 0.2, 0.3, 0.4, 0.5\}$, and $m$ values of $\{8, 16, 32\}$. The order of the time-variant convolutional filter $K$ is equal to the maximum sequence length $N$, which is set to 50. The batch size is set to 256, the dimension $D$ to 64, and the number of time-variant blocks $L$ to 2. We use the Adam optimizer for training. For reproducibility, the best hyperparameter settings are detailed in Table 7.

Table 7: Best hyperparameters of TV-Rec.

| Dataset | Learning Rate | $\alpha$ | $p$ | $m$ |
|---|---|---|---|---|
| | $L = 50$ | | | |
| Beauty | $5 \times 10^{-4}$ | $0$ | 0.5 | 32 |
| Sports | $5 \times 10^{-4}$ | $0$ | 0.5 | 16 |
| Yelp | $5 \times 10^{-4}$ | $1 \times 10^{-3}$ | 0.1 | 16 |
| LastFM | $1 \times 10^{-3}$ | $1 \times 10^{-3}$ | 0.4 | 8 |
| ML-1M | $1 \times 10^{-3}$ | $1 \times 10^{-5}$ | 0.3 | 8 |
| Foursquare | $5 \times 10^{-4}$ | $1 \times 10^{-5}$ | 0.2 | 8 |
| | $L = 200$ | | | |
| ML-1M | $1 \times 10^{-3}$ | $0$ | 0.1 | 16 |
| Foursquare | $5 \times 10^{-4}$ | $1 \times 10^{-5}$ | 0.1 | 8 |

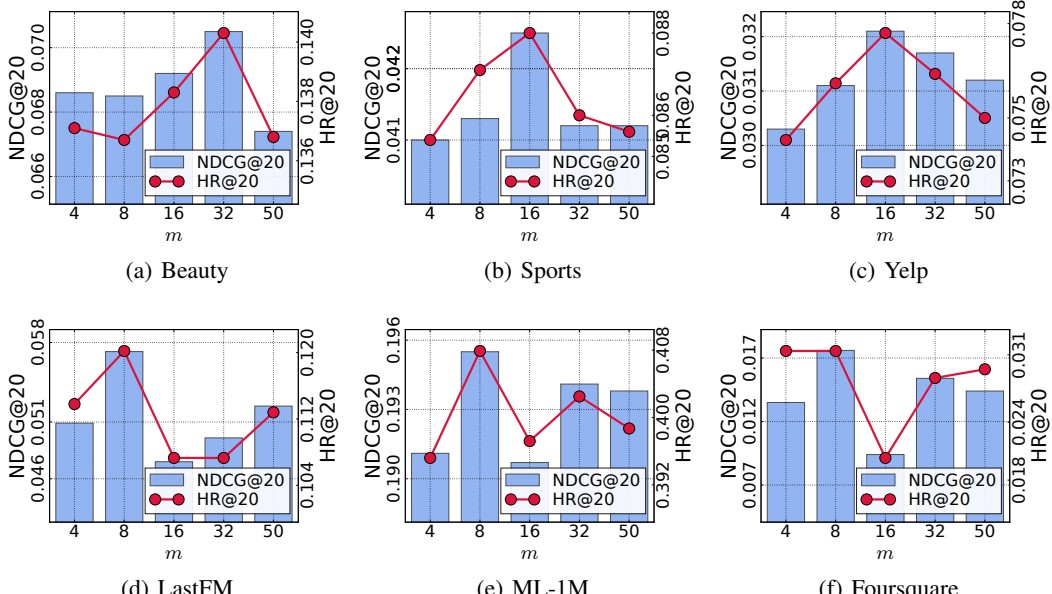

Figure 8: Sensitivity to the number of basis vectors $m$.

## E  Sensitivity Studies

In this section, we investigate the sensitivity of four hyperparameters: the number of basis vector $m$, dropout rate $p$, orthogonal regularization coefficient $\alpha$ and filter order $K$. The results are shown in Figs. 8, 9, 10 and Table 8 respectively. We keep optimal settings for all other hyperparameters except the one being examined.

**Sensitivity to $m$.**  In our time-variant graph filter, we use a basis matrix $\mathbf{B} \in \mathbb{R}^{m \times (K+1)}$ as shown in Eq. (8). The parameter $m$ determines the number of basis vectors, which affects the expressive power and computational complexity. To explore its effect, we extend the initial parameter search range of $m = \{8, 16, 32\}$ to include $m = \{4, 50\}$. Fig. 8 shows NDCG@20 and HR@20 by varying $m$. For Beauty, the best accuracy is achieved with $m = 32$, and both NDCG@20 and HR@20 drop significantly as $m$ decreases. This suggests that for Beauty, a larger number of basis vectors is beneficial, possibly due to the complexity of user-item interactions in this dataset. In contrast, for LastFM, the best accuracy is achieved with $m = 8$, and as $m$ increases, both NDCG@20 and HR@20 decline dramatically. This behavior may be attributed to the particular characteristics of music recommendation on LastFM, where a more condensed representation is sufficient to capture user preferences.

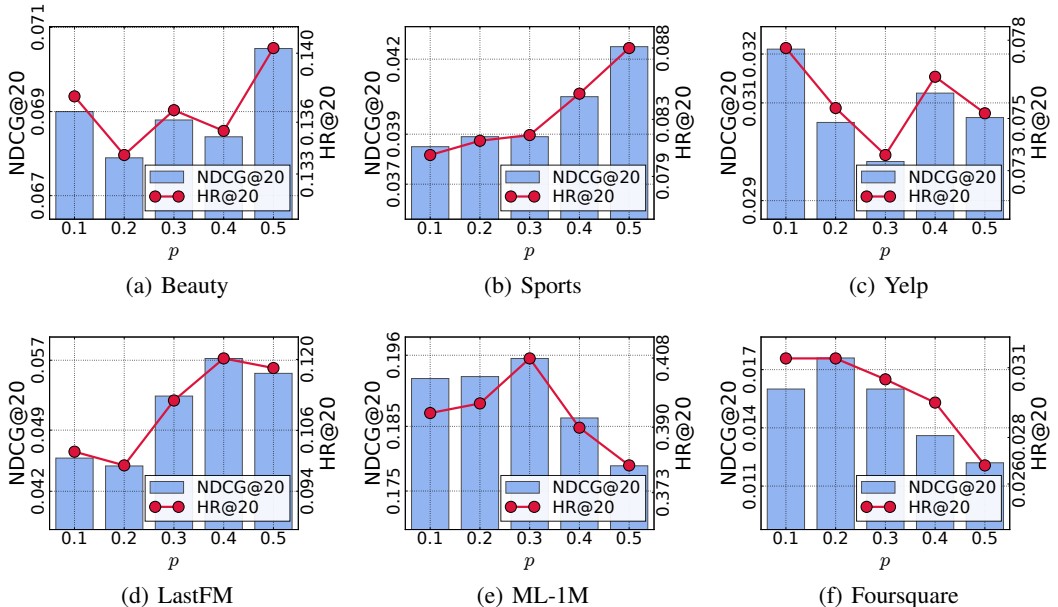

Figure 9: Sensitivity to dropout rate $p$.

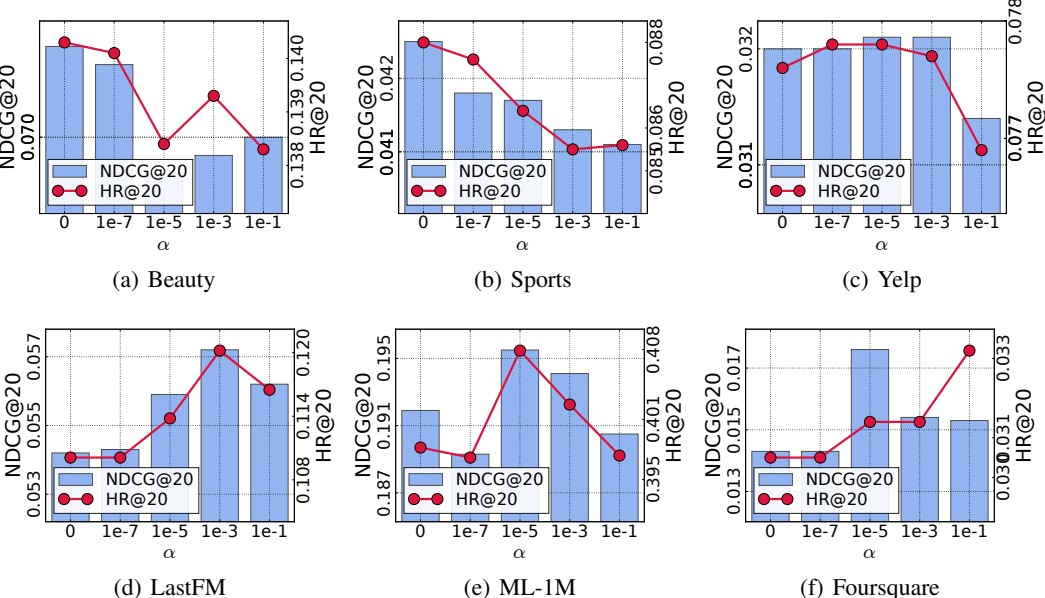

Figure 10: Sensitivity to orthogonal regularization coefficient $\alpha$.

**Sensitivity to $p$.** The effect of the dropout rate $p$ is analyzed in Fig. 9. For Sports, a larger value of $p$ leads to better performance. Conversely, for Foursquare, a lower value of $p$ results in improved performance. Datasets with many interactions tend to achieve better accuracy with a small $p$, whereas those with fewer interactions typically perform better with a large $p$. This is because lower data diversity of datasets with fewer interactions leads model to easily overfit. A higher dropout rate helps prevent overfitting by encouraging the model to learn more general patterns.

**Sensitivity to $\alpha$.** The parameter search range for $\alpha$ is extended from $\{0, 1 \times 10^{-3}, 1 \times 10^{-5}\}$ to $\{0, 1 \times 10^{-1}, 1 \times 10^{-3}, 1 \times 10^{-5}, 1 \times 10^{-7}\}$, and experiments are conducted with these values. Fig. 10 shows NDCG@20 and HR@20 by varying $\alpha$. For LastFM, the best accuracy is achieved

Table 8: Sensitivity to filter order $K$.

| $K$ | Beauty | | Sports | | Yelp | | LastFM | | ML-1M | | Foursquare | |
|---|---|---|---|---|---|---|---|---|---|---|---|---|
| | H@20 | N@20 | H@20 | N@20 | H@20 | N@20 | H@20 | N@20 | H@20 | N@20 | H@20 | N@20 |
| 3 | 0.1380 | 0.0690 | 0.0839 | 0.0405 | 0.0731 | 0.0295 | 0.1046 | 0.0443 | 0.3901 | 0.1843 | 0.0240 | 0.0116 |
| 5 | 0.1364 | 0.0680 | 0.0849 | 0.0407 | 0.0752 | 0.0308 | 0.1147 | 0.0490 | 0.3823 | 0.1814 | 0.0268 | 0.0116 |
| 10 | 0.1378 | 0.0688 | 0.0853 | 0.0410 | 0.0741 | 0.0303 | 0.1165 | 0.0532 | 0.3922 | 0.1898 | 0.0268 | 0.0139 |
| 25 | 0.1371 | 0.0684 | 0.0854 | 0.0409 | 0.0774 | **0.0323** | **0.1248** | 0.0546 | 0.3957 | 0.1891 | 0.0295 | 0.0163 |
| 50 | **0.1403** | **0.0705** | **0.0880** | **0.0425** | **0.0777** | 0.0321 | 0.1202 | **0.0572** | **0.4079** | **0.1955** | **0.0314** | **0.0176** |

Table 9: Results on XLong.

| Metric | SASRec | LRURec | TV-Rec | Improv. |
|---|---|---|---|---|
| HR@5 | 0.3612 | 0.4266 | **0.4844** | 13.55% |
| HR@10 | 0.4680 | 0.5137 | **0.5353** | 4.20% |
| HR@20 | 0.5612 | **0.5874** | 0.5774 | -1.70% |
| NDCG@5 | 0.2656 | 0.3227 | **0.3905** | 21.00% |
| NDCG@10 | 0.2979 | 0.3510 | **0.4071** | 15.98% |
| NDCG@20 | 0.3232 | 0.3697 | **0.4178** | 13.01% |

with $\alpha = 10^{-3}$. Both NDCG@20 and HR@20 drop significantly as $\alpha$ decreases, suggesting that an adequate level of orthogonal regularization is crucial to maintain effective filter diversity and prevent overfitting in this dataset. In contrast, for Sports and Yelp datasets, the highest accuracy occurs at relatively lower $\alpha$ values, and performance deteriorates noticeably as $\alpha$ increases. This behavior may indicate that too strong orthogonal regularization overly constrains the filter parameters, limiting the model's ability to adapt to the data distribution in these domains. These results emphasize the importance of adjusting $\alpha$ based on the characteristics of the dataset, balancing the regularization strength to achieve optimal performance.

**Sensitivity to filter order $K$.** We analyze the sensitivity of the filter order $K$, which is equivalent to the kernel size in CNN-based models. As shown in Table 8, except for Yelp and LastFM, setting $K = 50$ consistently yields the best performance across datasets. This supports our hypothesis that aligning the shift depth $K$ with the sequence length $N$ enables the model to capture more global context, thereby enhancing recommendation quality.

# F   Additional Results on XLong

To further test scalability in extreme cases, we conducted experiments on the XLong dataset, which contains 69,069 users, 2.12 million items, and approximately 66.8 million interactions. The sequences are exceptionally long, with an average length of 958.8 and a density of about $5 \times 10^{-4}$, roughly 20 times longer than in our main experiments. We followed the experimental settings of LRURec [43] and performed the experiments within the LRURec framework. As shown in Table 9, although TVRec shows slightly lower performance compared to LRURec in Recall@20, it significantly outperforms in all other metrics. This demonstrates that TVRec handles extremely long sequences effectively, highlighting its strength in long-range modeling tasks.

# G   Model Complexity and Runtime Analyses

In this section, we provide a comprehensive analysis of model complexity and runtime efficiency for the entire datasets. Table 10 shows a detailed comparison of our proposed TV-Rec model with 7 baseline models in 6 different datasets. Our TV-Rec consistently shows competitive parameter efficiency in all datasets, maintaining a comparable or slightly lower number of parameters than most advanced baselines. In terms of training efficiency, our TV-Rec shows competitive training cost in most datasets, often comparable to or slightly higher than SASRec and FMLPRec. TV-Rec shows strong inference efficiency, often having the lowest or near-lowest inference costs among all models, especially in datasets such as Beauty, Sports, and Yelp.

Table 10: Parameters number and execution efficiency analysis of models.

| Dataset | Metrics | TV-Rec | AdaMCT | BSARec | FMLPRec | NextItNet | BERT4Rec | SASRec | Caser |
|---|---|---|---|---|---|---|---|---|---|
| Beauty | # Parameters | 854,208 | 878,208 | 880,318 | 851,200 | 981,696 | 877,888 | 877,824 | 2,909,532 |
| | Training Cost (s/epoch) | 13.20 | 12.30 | 11.35 | 11.85 | 18.9184 | 21.98 | 11.21 | 65.54 |
| | Inference Cost (s/epoch) | 0.5697 | 0.6647 | 0.7299 | 0.5427 | 1.0267 | 0.5821 | 0.6316 | 1.0641 |
| | NDCG@20 | 0.0704 | 0.0691 | 0.0664 | 0.0419 | 0.0547 | 0.0484 | 0.0379 | 0.0164 |
| Sports | # Parameters | 1,248,160 | 1,278,592 | 1,264,318 | 1,251,584 | 1,644,224 | 1,278,272 | 1,278,208 | 4,835,756 |
| | Training Cost (s/epoch) | 19.46 | 17.63 | 18.44 | 17.76 | 30.1034 | 31.60 | 14.26 | 98.21 |
| | Inference Cost (s/epoch) | 0.7411 | 0.9049 | 0.9679 | 0.7049 | 1.2724 | 0.8016 | 0.8460 | 1.5715 |
| | NDCG@20 | 0.0428 | 0.0422 | 0.0379 | 0.0239 | 0.0316 | 0.0288 | 0.0217 | 0.0109 |
| Yelp | # Parameters | 1,355,424 | 1,385,856 | 1,365,522 | 1,358,848 | 1,751,488 | 1,385,536 | 1,385,472 | 3,925,238 |
| | Training Cost (s/epoch) | 21.89 | 20.87 | 21.83 | 20.20 | 32.88 | 37.57 | 16.74 | 111.62 |
| | Inference Cost (s/epoch) | 0.6388 | 0.8527 | 0.8890 | 0.6223 | 1.2348 | 0.7225 | 0.7322 | 1.3220 |
| | NDCG@20 | 0.0338 | 0.0290 | 0.0272 | 0.0211 | 0.0276 | 0.0275 | 0.0180 | 0.0151 |
| LastFM | # Parameters | 303,440 | 337,088 | 322,814 | 310,080 | 981,696 | 336,768 | 336,704 | 998,646 |
| | Training Cost (s/epoch) | 2.41 | 2.51 | 2.66 | 2.51 | 19.1253 | 4.29 | 2.30 | 13.22 |
| | Inference Cost (s/epoch) | 0.2649 | 0.2807 | 0.3228 | 0.2678 | 1.068 | 0.3018 | 0.3061 | 0.3445 |
| | NDCG@20 | 0.0582 | 0.0514 | 0.0485 | 0.0475 | 0.0402 | 0.0366 | 0.0458 | 0.0306 |
| ML-1M | # Parameters | 298,368 | 322,368 | 308,094 | 295,360 | 556,928 | 322,048 | 321,984 | 961,326 |
| | Training Cost (s/epoch) | 26.60 | 27.68 | 31.40 | 24.67 | 40.7731 | 46.51 | 22.50 | 102.99 |
| | Inference Cost (s/epoch) | 0.4129 | 0.4180 | 0.4474 | 0.3594 | 0.7269 | 0.4202 | 0.4181 | 0.5817 |
| | NDCG@20 | 0.1951 | 0.1836 | 0.1711 | 0.1388 | 0.1829 | 0.1588 | 0.1438 | 0.1101 |
| Foursquare | # Parameters | 719,040 | 743,040 | 728,766 | 716,032 | 1,108,672 | 742,720 | 742,656 | 1,073,044 |
| | Training Cost (s/epoch) | 6.01 | 5.79 | 5.81 | 5.11 | 8.9109 | 9.06 | 5.27 | 21.01 |
| | Inference Cost (s/epoch) | 0.2900 | 0.2913 | 0.3562 | 0.2521 | 0.4963 | 0.2871 | 0.3178 | 0.3703 |
| | NDCG@20 | 0.0170 | 0.0147 | 0.0131 | 0.0110 | 0.0115 | 0.0132 | 0.0137 | 0.0133 |

Table 11: Performance comparison between TV-Rec and the second-best baseline methods across 6 datasets. Results show the mean and standard deviation for 10 runs with different random seeds using the best hyperparameter settings.

| Datasets | Methods | HR@5 | HR@10 | HR@20 | NDCG@5 | NDCG@10 | NDCG@20 |
|---|---|---|---|---|---|---|---|
| Beauty | BSARec | 0.0694±0.001 | 0.0978±0.002 | 0.1352±0.002 | 0.0496±0.001 | 0.0587±0.001 | 0.0681±0.001 |
| | TV-Rec | 0.0706±0.001 | 0.0997±0.001 | 0.1375±0.002 | 0.0500±0.001 | 0.0594±0.001 | 0.0689±0.001 |
| Sports | BSARec | 0.0417±0.001 | 0.0600±0.001 | 0.0844±0.001 | 0.0288±0.001 | 0.0349±0.001 | 0.0411±0.001 |
| | TV-Rec | 0.0420±0.001 | 0.0610±0.002 | 0.0863±0.002 | 0.0290±0.001 | 0.0351±0.001 | 0.0415±0.001 |
| Yelp | DuoRec | 0.0268±0.001 | 0.0453±0.001 | 0.0733±0.001 | 0.0170±0.000 | 0.0230±0.000 | 0.0300±0.000 |
| | TV-Rec | 0.0284±0.001 | 0.0472±0.001 | 0.0759±0.001 | 0.0179±0.000 | 0.0240±0.001 | 0.0312±0.001 |
| LastFM | BSARec | 0.0501±0.004 | 0.0707±0.006 | 0.1051±0.008 | 0.0342±0.002 | 0.0412±0.002 | 0.0498±0.002 |
| | TV-Rec | 0.0508±0.005 | 0.0750±0.006 | 0.1090±0.007 | 0.0343±0.003 | 0.0420±0.003 | 0.0506±0.003 |
| ML-1M | LRURec | 0.1955±0.002 | 0.2818±0.002 | 0.3871±0.002 | 0.1326±0.002 | 0.1604±0.002 | 0.1869±0.002 |
| | TV-Rec | 0.2024±0.005 | 0.2901±0.005 | 0.3972±0.005 | 0.1365±0.004 | 0.1647±0.003 | 0.1918±0.003 |
| Foursquare | BSARec | 0.0133±0.003 | 0.0175±0.003 | 0.0250±0.003 | 0.0098±0.002 | 0.0111±0.002 | 0.0130±0.002 |
| | TV-Rec | 0.0151±0.002 | 0.0214±0.003 | 0.0289±0.002 | 0.0105±0.002 | 0.0126±0.002 | 0.0145±0.002 |

Overall, TV-Rec presents a balance between model complexity, computational efficiency, and recommendation performance. Our TV-Rec achieves the best performance metrics while maintaining competitive or better efficiency in both training and inference compared to state-of-the-art models.

# H   Statistical Significance of Experimental Results

To ensure the reliability of our evaluation, we conducted each experiment using 10 different random seeds under the best hyperparameter settings for both our proposed model and the second-best baseline model. The second-best models for each dataset are as follows: BSARec for Beauty, Sports, LastFM, and Foursquare; DuoRec for Yelp; and LRURec for ML-1M. We then report the mean and standard deviation of the performance metrics calculated across these runs to reflect variability caused by random initialization. The detailed results, including these statistics, are provided in Table 11.

# I    Broader Impact

This work proposes TV-Rec, a time-variant convolutional filter for sequential recommendation. Its improved efficiency and performance can positively impact real-world recommender systems by reducing computational cost and energy consumption. However, potential negative societal impacts include reinforcing user biases, limiting content diversity, and risking privacy violations if sensitive user behavior data is mishandled. TV-Rec's reliance on user interaction logs raises concerns about data privacy and fairness in recommendations. To mitigate these issues, we recommend careful data governance, fairness auditing, and the development of privacy-preserving training pipelines.

# J    Limitation

While TV-Rec achieves strong performance and inference-time efficiency, it has several limitations. First, the training process of TV-Rec involves some additional computational complexity due to repeated GFT and inverse GFT operations, as well as the generation of position-specific filters. Moreover, to enable spectral filtering, TV-Rec replaces the original line graph with DCG, which adds extra nodes through padding and increases the dimensionality of the spectral domain. These factors can lead to moderate increases in training time and memory usage. However, as shown in Table 10, this overhead remains manageable and does not significantly impact training scalability in practice. Second, the filter generation in TV-Rec is data-independent, relying solely on temporal position rather than sequence content. While this design improves generalization and structural simplicity, it may limit the model's adaptability to instance-specific behavior or irregular patterns. Nonetheless, the inference-time efficiency and architectural simplicity make these trade-offs acceptable for many practical applications. Overall, these trade-offs are justified by TV-Rec's strong empirical results and practical efficiency.

