# OpenReview forum: "TV-Rec: Time-Variant Convolutional Filter for Sequential Recommendation"
_NeurIPS.cc/2025/Conference — NeurIPS 2025 poster_

### Official Review · Reviewer_ANZm · 2025-06-05

**Clarity:** 3
**Significance:** 3
**Originality:** 3
**Rating:** 4
**Confidence:** 2

**Summary:**

The paper proposes a new technique to allow sequential recommendation models to more flexibly adapt to different temporal patterns. Existing sequential models are fairly "rigid" (just an ordered sequence of events), whereas the proposed approach can potentially be more adaptive to context and different dynamics at different positions in the sequence. The specific contribution is based on a time-varying convolutional filter.

**Questions:**

-- Can you comment on model scalibility? I see there is some scalability analysis, though the method seems not to have been evaluated on large datasets.

-- The results in the scalability table don't seem to match the numbers in the table (FMLPRec seems to be the winner in the figure?)

-- I don't totally follow what's shown in the inference time analysis. Many of these methods reduce inference to nearest-neighbor search in some latent space such that I don't really see why inference time should vary so much (though maybe that's sort of what the table shows). Though might have misunderstood something here.

**Ethical Concerns:**

["NO or VERY MINOR ethics concerns only"]

**Final Justification:**

The clarifications made by the authors seem reasonable. They do add new experiments which go beyond normal "clarifications" though I don't mind that too much. Otherwise my scores and opinions seem not too different from other reviewers (and also borderline) so am keeping the score.

**Limitations:**

yes

**Quality:**

3

**Strengths And Weaknesses:**

-- Reasonable idea which seems to offer a unique perspective on a problem that's studied in a large number of papers.

-- Fairly promising results.

-- A decent enough set of baselines and datasets, to the point that the experiments are reasonably thorough.

-- Datasets maybe on the small side, not sure about scalability.

---

> ### Author Rebuttal · Authors · 2025-07-31
>
> Thank you for your thoughtful and constructive feedback on our work. Below, we provide detailed responses to each of your concerns, including additional experiments and revisions that we will incorporate into the final version of the paper.
>
> ## **1. Concerns on Dataset Scale and Model Scalability**
> In sequential recommendation, scalability is often discussed with repect to sequence length. While our main experiments use a commonly adopted maximum sequence length of 50, we have included additional results with a much longer length of 200 (4 times longer) in Table 3 and Section 4.1.
>
> Additionally, to further test scalability in extreme cases, we conducted experiments on the XLong dataset with a maximum sequence length of 1,000--20 times longer than in our main experiment. The dataset statistics are summarized below:
>
> **Table D. Statistics of the XLong Dataset**
> |Dataset|XLong|
> |-|-|
> |\# of Users|69,069|
> |\# of Items|2,122,932|
> |\# of Interactions|66.8M|
> |Average Length|958.8|
> |Density|5e-4|
>
> We followed the experimental settings of LRURec [1] and performed the experiments within the LRURec framework. The results are as follows:
>
> **Table E. Results on the XLong dataset**
>
> | Metric    | SASRec  | LRURec  | TV-Rec   | Improvement |
> |-|-|-|-|-|
> | Recall@5  | 0.3612  | 0.4266  | 0.4844  | 13.55% |
> | Recall@10 | 0.4608  | 0.5137  | 0.5353  | 4.20%  |
> | Recall@20 | 0.5612  | 0.5874  | 0.5774  | -1.70% |
> | NDCG@5    | 0.2656  | 0.3227  | 0.3905  | 21.00% |
> | NDCG@10   | 0.2979  | 0.3510  | 0.4071  | 15.98% |
> | NDCG@20   | 0.3232  | 0.3697  | 0.4178  | 13.01% |
>
> Although TVRec shows slightly lower performance compared to LRURec in Recall@20, it significantly outperforms in all other metrics. This demonstrates that TVRec handles extremely long sequences effectively, highlighting its strength in long-range modeling tasks.
>
> ## **2. Clarifying the purpose of Figure 5**
>
> Thank you for pointing this out. You're correct--FMLPRec achieves the fastest inference time, as reported in Appendix Table 7 and also depicted in Figure 5. However, the purpose of Figure 5 is not to emphasize inference speed alone, but rather to present a broader trade-off between inference time and recommendation accuracy (NDCG@20).
>
> While FMLPRec is computationally efficient, its recommendation accuracy is significantly lower than that of other hybrid models such as TV-Rec, BSARec, and AdaMCT. Therefore, our key intent in Figure 5 is to highlight how TV-Rec achieves an excellent balance between speed and accuracy, especially among high-performing models.
>
> We acknowledge that this intent might not have been sufficiently clear, and we will revise the caption and explanation in the main paper to emphasize that the figure aims to show the accuracy-efficiency trade-off, rather than showcasing the absolute fastest model.
>
> ## **3. What does inference time measure?**
>
> We would like to clarify that the inference time reported in Figure 5 and Table 7 refers to the computation time of the user sequence embedding, not the time taken for similarity-based retrieval or ranking over the item set.
>
> Our focus in this work is on designing an efficient and expressive sequence encoder that captures temporal dynamics in user behavior. All compared models in our study share similar post-encoding steps (e.g., computing dot-product scores with item embeddings), so the variation in inference time mainly stems from the complexity of the sequence encoding architecture.
>
> We will clarify this point in the main paper to ensure that the reported inference time is properly interpreted as the time for generating the sequence embedding.
>
>
> ## **Reference**
> [1] Linear Recurrent Units for Sequential Recommendation, WSDM 2024

---

### Official Review · Reviewer_xQNR · 2025-06-25

**Clarity:** 2
**Significance:** 2
**Originality:** 3
**Rating:** 4
**Confidence:** 4

**Summary:**

This paper proposes TV-Rec, a sequential recommendation model that leverages graph signal processing to perform spectral filtering on user behavior sequences. By constructing time-variant filters for each position via a low-rank formulation, the model avoids using position embeddings or attention. Experimental results on six datasets show strong performance and efficiency compared to prior methods.

**Questions:**

1.	What is the theoretical expressive capacity of the filter design?

Can the authors formalize whether the C \cdot \bar{B} construction is capable of approximating arbitrary frequency responses under certain norms or graph structures?

2.	Can personalization be incorporated into filters?

Currently, all filters are position-specific but shared across users. Could user- or item-aware adaptation (e.g., via embeddings or gating) enhance personalization, especially in cold-start scenarios?

3.	How does TV-Rec handle data sparsity or cold-start scenarios?

Given its dependence on spectral structures, how robust is the model when faced with sparse user interactions or new users/items with limited historical data, which are common challenges in real-world recommendation systems?

**Ethical Concerns:**

["NO or VERY MINOR ethics concerns only"]

**Final Justification:**

The author's rebuttal has addressed some of my concerns. Thus, I will increase the score.

**Quality:**

3

**Strengths And Weaknesses:**

Strengths:
1. The paper introduces a novel sequential recommendation approach using spectral graph signal processing (GSP). Its key innovation is time-variant convolutional filters assigning unique spectral filters per position, replacing position embeddings and self-attention with an efficient low-rank factorization.
2. Experiments on six benchmarks show consistent gains over 10 state-of-the-art models, supported by ablation studies and efficiency analysis.
3. The approach offers a compelling alternative to the dominant Transformer-based models, which often require heavy computation and explicit position encoding.

Weaknesses:
1. Key design choices such as the number of basis vectors m, shift depth K, or basis configuration ω are not systematically analyzed. It remains unclear whether the method is robust under different settings.
2. The basis matrix \bar{B} is constructed heuristically via Vandermonde form, but the selection of frequency anchors and the number of basis vectors lack theoretical or empirical justification. Please explain the rationale behind their choice in the main paper.
3. Strong recent models like Tim4Rec are not included in the comparison. Their absence weakens the empirical competitiveness of the claims.

---

> ### Author Rebuttal · Authors · 2025-07-31
>
> Thank you for your careful and insightful review of our manuscript. Below, we respond to each point with further details and experimental results.
>
> ## **1. Regarding absence of strong recent models**
> We could not find any official publication or preprint corresponding to Time4Rec (ICLR 2024). If you could kindly provide a reference or link to the paper, we would be happy to verify it and include it in our comparison.
>
> Additionally, we have incorporated results for several strong and recent graph-based models [1,2,3,4] in response to reviewer HzEw’s comments. We kindly invite you to refer to our response to reviewer HzEw (Answer 1) for the updated comparisons.
>
>
> ## **2. Expressive capacity or time-variant convolutional filter**
> Our proposed filter design can be viewed as a position-aware spectral filtering mechanism, where the filter matrix is constructed as: $\mathbf{G} = (\mathbf{U} \circ (\mathbf{H} \mathbf{\Lambda}^\top)) \mathbf{U}^\top,\quad \text{with } \mathbf{H} = \mathbf{C} \bar{\mathbf{B}}$. Here, $\mathbf{U}$ is the Graph Fourier Transform (GFT) matrix, $\bar{\mathbf{B}}$ is a truncated spectral basis, and $\mathbf{C}$ is a learnable coefficient matrix that modulates the frequency response in a position-dependent manner. This construction enables the model to express frequency-wise filters via linear combinations of the basis functions in $\bar{\mathbf{B}}$. The expressive capacity of the filter is thus determined by the span of $\bar{\mathbf{B}}$: any frequency response vector within this subspace can be exactly represented, and more general responses can be approximated in the $\ell_2$ sense as the number of basis vectors increases. Compared to fixed convolutional filters, our formulation allows more flexible spectral filtering. While we do not provide a formal approximation bound under a specific norm, the empirical results in Table 4 suggest that our construction is sufficiently expressive for modeling real-world sequential data. A theoretical characterization of the approximation capacity remains an interesting direction for future work.
>
> ## **3. Analysis on hyper-parameters**
> We would like to clarify that the basis matrix $B$ is not derived from a Vandermonde matrix, nor constructed using fixed frequency anchors. Instead, it is a fully learnable parameter matrix, jointly optimized during training. This allows the model to adaptively discover suitable spectral representations without relying on heuristic or hand-crafted frequency configurations.
> Regarding the number of basis vectors $m$, we have already included a sensitivity analysis in Appendix E.1, and we will update the main paper to explicitly refer to this study for clarity.
> As for the shift depth $K$, we initially set it equal to the sequence length $N$ following a full-coverage design. To address your concern, , we conducted an additional ablation varying $K$, and summarize the results in table below. As shown, except for Yelp, setting $K=50$ consistently yields the best performance across datasets. This supports our hypothesis that aligning the shift depth $K$ with the sequence length $N$ enables the model to capture more global context, thereby enhancing recommendation quality.
>
> **Table C. Ablation Study on $K$**
> | dataset    | K  | HR@05  | HR@10  | HR@20  | NDCG@05 | NDCG@10 | NDCG@20 |
> |------------|----|--------|--------|--------|---------|---------|---------|
> | Beauty     | 3  | 0.0704 | 0.0992 | 0.1380 | 0.0499  | 0.0592  | 0.0690  |
> |      | 5  | 0.0701 | 0.0996 | 0.1364 | 0.0492  | 0.0587  | 0.0680  |
> |      | 10 | 0.0703 | 0.0986 | 0.1378 | 0.0498  | 0.0589  | 0.0688  |
> |      | 25 | 0.0700 | 0.0982 | 0.1371 | 0.0495  | 0.0586  | 0.0684  |
> |      | 50 | 0.0721 | 0.1017 | 0.1403 | 0.0513  | 0.0608  | 0.0705  |
> | Sports     | 3  | 0.0408 | 0.0596 | 0.0839 | 0.0283  | 0.0344  | 0.0405  |
> |      | 5  | 0.0414 | 0.0601 | 0.0849 | 0.0285  | 0.0344  | 0.0407  |
> |      | 10 | 0.0415 | 0.0595 | 0.0853 | 0.0287  | 0.0345  | 0.0410  |
> |      | 25 | 0.0403 | 0.0592 | 0.0854 | 0.0283  | 0.0343  | 0.0409  |
> |      | 50 | 0.0431 | 0.0635 | 0.0880 | 0.0298  | 0.0363  | 0.0425  |
> | Yelp       | 3  | 0.0270 | 0.0458 | 0.0731 | 0.0166  | 0.0226  | 0.0295  |
> |        | 5  | 0.0272 | 0.0455 | 0.0752 | 0.0174  | 0.0233  | 0.0308  |
> |        | 10 | 0.0280 | 0.0465 | 0.0741 | 0.0174  | 0.0233  | 0.0303  |
> |        | 25 | 0.0290 | 0.0494 | 0.0774 | 0.0188  | 0.0253  | 0.0323  |
> |        | 50 | 0.0290 | 0.0474 | 0.0777 | 0.0186  | 0.0245  | 0.0321  |
> | LastFM     | 3  | 0.0413 | 0.0642 | 0.1046 | 0.0266  | 0.0340  | 0.0443  |
> |      | 5  | 0.0514 | 0.0780 | 0.1147 | 0.0315  | 0.0400  | 0.0490  |
> |      | 10 | 0.0523 | 0.0771 | 0.1165 | 0.0355  | 0.0434  | 0.0532  |
> |      | 25 | 0.0560 | 0.0862 | 0.1248 | 0.0352  | 0.0448  | 0.0546  |
> |      | 50 | 0.0596 | 0.0853 | 0.1202 | 0.0402  | 0.0484  | 0.0572  |
> | ML-1M      | 3  | 0.1907 | 0.2778 | 0.3901 | 0.1281  | 0.1562  | 0.1843  |
> |       | 5  | 0.1874 | 0.2768 | 0.3823 | 0.1260  | 0.1547  | 0.1814  |
> |       | 10 | 0.2000 | 0.2879 | 0.3922 | 0.1354  | 0.1636  | 0.1898  |
> |       | 25 | 0.1960 | 0.2844 | 0.3957 | 0.1325  | 0.1611  | 0.1891  |
> |       | 50 | 0.2013 | 0.2904 | 0.4079 | 0.1371  | 0.1658  | 0.1955  |
> | Foursquare | 3  | 0.0120 | 0.0194 | 0.0240 | 0.0081  | 0.0105  | 0.0116  |
> |  | 5  | 0.0129 | 0.0203 | 0.0268 | 0.0075  | 0.0099  | 0.0116  |
> |  | 10 | 0.0148 | 0.0166 | 0.0268 | 0.0109  | 0.0115  | 0.0139  |
> |  | 25 | 0.0222 | 0.0249 | 0.0295 | 0.0142  | 0.0151  | 0.0163  |
> |  | 50 | 0.0175 | 0.0259 | 0.0314 | 0.0134  | 0.0161  | 0.0176  |
>
>
>
> ## **4. Can personalization be incorporated into filters?**
> As discussed in Appendix I, TV-Rec is data-independent, meaning that it does not incorporate user- or item-specific information into filter generation. While this enables lower complexity and efficient inference, it also limits the model's ability to perform fine-grained personalization, such as user-specific self-attention in Transformer-based models. Incorporating user- or item-aware filters--while preserving the efficiency benefits of our graph-based framework--is an important direction for future work.
>
> ## **5. How does TV-Rec handle data sparsity or cold-start scenarios?**
> First, we note that our experiments are already conducted on datasets with sparse user-item interactions, which reflects realistic recommendation scenarios.
> Regarding the cold-start problem, TV-Rec, like most sequential recommendation models, relies on learned item embeddings from historical interactions. As a result, it cannot directly handle entirely new (unseen) items without interaction history. Nonetheless, if side information such as item attributes or metadata is available, the model could be naturally extended to support new items by generating embeddings from such auxiliary features.
> For new users, since sequential recommendation models inherently rely on sequences of interactions, it is difficult to make personalized predictions when no interaction history is available at all. However, if a user has even 1–2 interactions, TV-Rec can form a valid input sequence and generate meaningful recommendations.
>
> ## Reference
> [1] Session-Based Recommendation with Graph Neural Networks, AAAI 2019
>
> [2] Graph Contextualized Self-Attention Network for Session-based Recommendation, IJCAI 2019
>
> [3] Enhancing Sequential Recommendation with Graph Contrastive Learning, IJCAI 2022
>
> [4] Self-supervised Graph Neural Sequential Recommendation with Disentangling Long and Short-Term Interest, ACM Transactions on Recommender Systems 2025

---

> > ### Comment · Reviewer_xQNR · 2025-08-04
> >
> > Thank you for the detailed responses regarding the hyperparameter experiments.
> >
> > 1. First, I apologize and need to clarify my earlier comment about "Tim4Rec (ICLR 2024)" - I was referring to this paper, which shares many datasets and baselines with the authors' work, but this paper was not published in ICLR. This was a citation error:
> > [1] Fan H, Zhu M, Hu Y, et al. Tim4rec: An efficient sequential recommendation model based on time-aware structured state space duality model[J]. arXiv preprint arXiv:2409.16182, 2024.
> >
> > 2. I noticed the authors' response to HzEw regarding the supplementary baselines. In the original LS4SRec paper, LS4SRec consistently outperforms GCL4SR, but in the authors' reported results, GCL4SR performs better than LS4SRec. Could the authors discuss this discrepancy?
> >
> > 3. My question about personalization wasn't about incorporating external content features, but rather whether the filter modeling layer should consider user/item-specific embeddings. Currently, all filters are shared across users - could this approach neglect sparse users/items in the typical long-tail distribution? Would incorporating user/item-specific attention in the filter design improve personalization without significantly increasing time complexity?
> >
> > 4. While the authors note that datasets are generally sparse, this observation may not fully address the original concern. Users and items within datasets exhibit varying interaction frequencies, creating different sparsity levels. A performance breakdown by grouping users/items according to their interaction counts would provide clearer insights into how the method performs across these different sparsity levels compared to baselines.

---

> > > ### Author Response · Authors · 2025-08-08
> > >
> > > We sincerely thank the reviewer xQNR for the helpful follow-up questions and clarifications. Please find our responses and experimental results below:
> > >
> > > ## **1. Additional baselines**
> > > Thank you for the clarification regarding the reference. We have completed experiments on the baseline you mentioned. Please refer to Table G for the complete results, along with additional GNN-based baselines. For Tim4Rec, we were unable to run experiments on the Yelp dataset because the required timestamps were not available in the preprocessed data we used. As shown in Table G, TV-Rec consistently outperforms the additional baselines across all six datasets, except for HR@10 on Yelp, further demonstrating the effectiveness of our time-variant filter design.
> > >
> > > **Table G. Comparison to additional methods**
> > >
> > > | Dataset | Metric   | SR-GNN | GC-SAN | GCL4SR | LS4SRec | Tim4Rec | TV-Rec |
> > > |-|-|-|-|-|-|-|-|
> > > | **Beauty**  | HR@5  | 0.0365  | 0.0570  | 0.0631 | 0.0626   |0.0639| 0.0721  |
> > > || HR@10 | 0.0567  | 0.0784  | 0.0880 | 0.0876   |0.0884| 0.1017  |
> > > || HR@20 | 0.0847  | 0.1059  | 0.1206 | 0.1200   |0.1220| 0.1403  |
> > > || NDCG@5    | 0.0238  | 0.0408  | 0.0438 | 0.0442   |0.0447| 0.0513  |
> > > || NDCG@10   | 0.0303  | 0.0477  | 0.0519 | 0.0522   |0.0526| 0.0608  |
> > > || NDCG@20   | 0.0374  | 0.0546  | 0.0601 | 0.0604   |0.0611| 0.0705  |
> > > | **Sports**  | HR@5    | 0.0210 | 0.0295 | 0.0360 | 0.0353  |0.0343| 0.0431 |
> > > |   | HR@10    | 0.0336 | 0.0423 | 0.0516 | 0.0516  |0.0487| 0.0635 |
> > > |   | HR@20    | 0.0517 | 0.0608 | 0.0744 | 0.0758  |0.0707| 0.0880 |
> > > |   | NDCG@05  | 0.0138 | 0.0201 | 0.0248 | 0.0244  |0.0239| 0.0298 |
> > > |   | NDCG@10  | 0.0179 | 0.0242 | 0.0298 | 0.0296  |0.0286| 0.0363 |
> > > |   | NDCG@20  | 0.0224 | 0.0289 | 0.0356 | 0.0357  |0.0341| 0.0425 |
> > > | **Yelp**    | HR@5    | 0.0225 | 0.0236 | 0.0248 | 0.0278  |-| 0.0290 |
> > > |    | HR@10    | 0.0377 | 0.0385 | 0.0417 | 0.0476  |-| 0.0474 |
> > > |    | HR@20    | 0.0609 | 0.0635 | 0.0684 | 0.0776  |-| 0.0777 |
> > > |    | NDCG@05  | 0.0145 | 0.0149 | 0.0155 | 0.0175  |-| 0.0186 |
> > > |    | NDCG@10  | 0.0194 | 0.0197 | 0.0209 | 0.0238  |-| 0.0245 |
> > > |    | NDCG@20  | 0.0252 | 0.0260 | 0.0276 | 0.0314  |-| 0.0321 |
> > > | **LastFM**  | HR@5  | 0.0385  | 0.0376  | 0.0385 | 0.0505   |0.0413| 0.0596  |
> > > |             | HR@10 | 0.0532  | 0.0569  | 0.0633 | 0.0752   |0.0670| 0.0853  |
> > > |             | HR@20 | 0.0872  | 0.0807  | 0.0908 | 0.1147   |0.0936| 0.1202  |
> > > |             | NDCG@5    | 0.0248  | 0.0273  | 0.0249 | 0.0352   |0.0305| 0.0402  |
> > > |             | NDCG@10   | 0.0295  | 0.0334  | 0.0328 | 0.0431  |0.0388| 0.0484  |
> > > |             | NDCG@20   | 0.0379  | 0.0394  | 0.0398 | 0.0531   |0.0455| 0.0572  |
> > > | **ML-1M**   | HR@5    | 0.1439 | 0.1724 | 0.1667 | 0.1758  |0.1851| 0.2013 |
> > > |   | HR@10    | 0.2070 | 0.2449 | 0.2470 | 0.2579  |0.2695| 0.2904 |
> > > |   | HR@20    | 0.2940 | 0.3255 | 0.3381 | 0.3675  |0.3687| 0.4079 |
> > > |   | NDCG@05  | 0.0967 | 0.1173 | 0.1120 | 0.1195  |0.1281| 0.1371 |
> > > |   | NDCG@10  | 0.1169 | 0.1407 | 0.1379 | 0.1459  |0.1553| 0.1658 |
> > > |   | NDCG@20  | 0.1390 | 0.1611 | 0.1607 | 0.1735  |0.1803| 0.1955 |
> > > | **Foursquare** | HR@5  | 0.0157  | 0.0148  | 0.0148 | 0.0148   |0.0175| 0.0175  |
> > > |             | HR@10 | 0.0203  | 0.0175  | 0.0166 | 0.0157   |0.0231| 0.0259  |
> > > |             | HR@20 | 0.0222  | 0.0212  | 0.0185 | 0.0194   |0.0249| 0.0314  |
> > > |             | NDCG@5    | 0.0117  | 0.0114  | 0.0113 | 0.0124   |0.0121| 0.0134  |
> > > |             | NDCG@10   | 0.0132  | 0.0122  | 0.0119 | 0.0127   |0.0139| 0.0161  |
> > > |             | NDCG@20   | 0.0137  | 0.0131  | 0.0123 | 0.0136   |0.0144| 0.0176  |
> > >
> > > ## **2. Discrepancy on the results (GCL4SR vs. LS4SRec)**
> > >
> > > Thank you for pointing this out. The results in Table G are obtained under the TV-Rec evaluation framework, which differs from the experimental setting used in the original LS4SRec paper. None of the datasets overlap, and the training protocols and computational resources also differ, making direct comparisons difficult.
> > >
> > > In our results, GCL4SR slightly outperforms LS4SRec on a few metrics in three datasets (Beauty, Sports, and Foursquare), but these gains are marginal and limited to specific metrics. In contrast, LS4SRec demonstrates consistently stronger performance across the remaining datasets and outperforms GCL4SR in most metrics overall. This suggests that the performance gap observed in our setting may be influenced by differences in data characteristics.
> > >
> > > Overall, our findings do not contradict the original LS4SRec paper, but rather highlight how performance may vary under different experimental settings.

---

> > > ### Author Response · Authors · 2025-08-08
> > >
> > > ## **3. Regarding personalization**
> > >
> > > Thank you for your clarification on personalization. While our current filter design shares parameters across users for efficiency, we agree that incorporating user- or item-specific adaptation could further improve performance, especially for long-tail users or in cold-start scenarios.
> > >
> > > We believe there are promising directions to explore personalization while maintaining computational efficiency:
> > >
> > > 1. **Output modulation:** Rather than modifying the filter directly, the output can be modulated using sequence- or user-level embeddings, through mechanisms such as gating or residual adaptation.
> > > 2. **Personalized filter coefficients:** The core part of our filter degisn is constructed as $H=C\bar{B}$, where $C$ is a coefficient matrix that varies across positions, and $\bar{B}$ is a shared basis. By conditioning $C$ on user embeddings (e.g., $C=MLP(e_u)$), the filter can be adapted per user while keeping the basis shared. Since $m << n$, the added complexity $O(nmd)$ remains small compared to the total cost $O(n^2m+n^2d)$, and substantially lower than that of self-attention layers, which require $O(nd^2+n^2d)$. This formulation also allows a flexible trade-off between personalization strength and computational cost by tuning $m$.
> > >
> > > While such personalization may complicate full precomputation of filters, our method would still retain a clear efficiency advantage. We consider this a promising direction for future work and believe it could complement the current design effectively.
> > >
> > > ## **4. Regarding user/item sparsity**
> > >
> > > Thank you for the suggestion to analyze performance across different sparsity levels. We conducted an additional breakdown of results using Beauty dataset by grouping users and items according to their interaction counts, where a user’s count denotes the length of their historical interaction sequence and an item’s count denotes the number of users who purchased it.
> > >
> > > As shown in Table H and I, performance gains are evident across all groups, with consistent improvements observed in sparse groups, indicating the method’s robustness to data sparsity.
> > >
> > > **Table H. Performance comparison across item interaction count groups**
> > >
> > > | # of interaction    | Metric   | AdaMCT  | BSARec  | TV-Rec  |
> > > |---------------------|----------|---------|---------|---------|
> > > | 0-5 (7%)            | HR@10    | 0.0317  | 0.0270  | 0.0350  |
> > > |                     | NDCG@10  | 0.0226  | 0.0198  | 0.0234  |
> > > | 6-15 (35%)          | HR@10    | 0.0391  | 0.0419  | 0.0436  |
> > > |                     | NDCG@10  | 0.0257  | 0.0252  | 0.0265  |
> > > | 16-40 (29%)         | HR@10    | 0.0790  | 0.0814  | 0.0867  |
> > > |                     | NDCG@10  | 0.0501  | 0.0503  | 0.0533  |
> > > | over 40 (29%)       | HR@10    | 0.1839  | 0.2016  | 0.2017  |
> > > |                     | NDCG@10  | 0.1091  | 0.1172  | 0.1181  |
> > > | All (100%) | HR@10 | 0.0925 | 0.0990 | 0.1017 |
> > > | | NDCG@10 | 0.0569 | 0.0590 | 0.0608 |
> > >
> > > **Table I. Performance comparison across user interaction count groups**
> > >
> > > | # of interaction    | Metric   | AdaMCT  | BSARec  | TV-Rec  |
> > > |---------------------|----------|---------|---------|---------|
> > > | 0-5 (32%)           | HR@10    | 0.0787  | 0.0863  | 0.0873  |
> > > |                     | NDCG@10  | 0.0480  | 0.0511  | 0.0526  |
> > > | 6-8 (39%)           | HR@10    | 0.0871  | 0.0945  | 0.0977  |
> > > |                     | NDCG@10  | 0.0541  | 0.0563  | 0.0585  |
> > > | 9-15 (20%)          | HR@10    | 0.0914  | 0.0986  | 0.1002  |
> > > |                     | NDCG@10  | 0.0566  | 0.0576  | 0.0581  |
> > > | over 15 (9%)        | HR@10    | 0.1702  | 0.1676  | 0.1770  |
> > > |                     | NDCG@10  | 0.1034  | 0.1037  | 0.1080  |
> > > | All (100%) | HR@10 | 0.0925 | 0.0990 | 0.1017 |
> > > | | NDCG@10 | 0.0569 | 0.0590 | 0.0608 |

---

### Official Review · Reviewer_Y8Cz · 2025-06-25

**Clarity:** 3
**Significance:** 3
**Originality:** 3
**Rating:** 5
**Confidence:** 4

**Summary:**

The paper introduces TV-Rec, a novel architecture for sequential recommendation that leverages time-variant convolutional filters inspired by graph signal processing. These filters adapt dynamically to different temporal positions in a user’s interaction sequence, in contrast to traditional fixed convolutional filters or self-attention mechanisms. TV-Rec eliminates the need for self-attention and positional embeddings while providing better performance and computational efficiency. The model is evaluated on six benchmark datasets and consistently outperforms ten state-of-the-art methods.

**Questions:**

- In Eq. (13), what do $B_{\text{real}}$ and $B_{\text{imag}}$ represent, and why is orthogonal regularization applied to these components?
- In Fig. 3a, how is the basic graph filter learned? Is it derived from a baseline method or from TV-Rec with static (non-time-variant) filters?

**Ethical Concerns:**

["NO or VERY MINOR ethics concerns only"]

**Final Justification:**

After considering the authors' rebuttal and the discussion, I am satisfied that my main concern—lack of comparison with graph-based methods—has been adequately addressed through additional experiments. The new results strengthen the empirical validation, and I encourage the authors to include them in the final version. I did not identify any significant unresolved issues, and the discussion with other reviewers further reinforced the paper's strengths in methodology, novelty, and empirical results. Based on this, I have increased my score and recommend acceptance.

**Limitations:**

yes

**Paper Formatting Concerns:**

see weakness

**Quality:**

3

**Strengths And Weaknesses:**

### **Strengths**

- The paper is well-written and clearly structured. The motivation is well-articulated, and Table 1 effectively contextualizes the proposed method among existing approaches.
- The concept of a time-aware (time-variant) convolutional filter is novel and intuitively meaningful.
- The method is grounded in solid theoretical foundations, with a well-developed formulation based on graph signal processing.
- The interpretability of TV-Rec is enhanced by filter behavior analysis and an illustrative case study.
- The experimental evaluation is thorough, covering both qualitative and quantitative aspects. The authors also include comprehensive hyperparameter analysis in the appendix.

### **Weaknesses**

- While the paper focuses on leveraging GSP for sequential recommendation, the benefits of this framework are not thoroughly connected to the SR context. For instance, in Lines 106–109, the statement that GSP helps "capture complex dependencies" is too vague. The paper would benefit from a more precise explanation of how GSP-derived filters concretely improve sequential modeling.
- The decision to model user sequences as a line graph is not well-justified. Why choose a line graph over other graph structures? How sensitive is the performance to the graph construction method? This should be discussed more explicitly.
- Given that TV-Rec uses graph-based modeling, it would be helpful to relate and compare this work to existing GNN-based recommendation methods. Even a brief discussion or experimental comparison would strengthen the paper.
- Although the authors provide mean and standard deviation of experimental results in the appendix, it would improve transparency to report p-values or statistical significance directly in the main text.

---

> ### Author Rebuttal · Authors · 2025-07-31
>
> Thank you for your thoughtful and constructive feedback on our work. Below, we provide detailed responses to each of your concerns, including additional experiments and revisions that we will incorporate into the final version of the paper.
>
> ## **1. Justification for using GSP and the line graph structure in sequential recommendation**
>
> The core idea of TV-Rec is to model user sequences as graph signals and process them in the spectral domain using GFT. This spectral representation allows us to decompose sequential patterns into interpretable frequency components: high-frequency components reflect abrupt behavioral changes, while low-frequency components capture stable, smooth trends. Although the current manuscript does not explicitly associate specific frequency bands with behavioral categories (e.g., long-term vs. short-term preferences), TV-Rec employs position-specific spectral filters that modulate these components adaptively. This mechanism allows the model to dynamically emphasize or suppress different frequencies depending on the position in the sequence, enabling a fine-grained and temporally aware representation of user behavior. In contrast to attention-based or CNN-based SR models—which typically rely on fixed receptive fields or explicit positional encodings—TV-Rec directly operates in the spectral domain, offering a principled and flexible way to encode temporal inductive bias. A similar spectral perspective is adopted in BSARec [1], which applies GFT to model sequential signals in the frequency domain and modulates them using a learnable frequency rescaler.
>
> Our choice of the line graph structure is motivated by both theoretical and practical considerations. First, designing GSP-based filters requires eigendecomposition of the graph Laplacian, which necessitates a normalized and consistent graph topology. The line graph satisfies this requirement and provides a uniform spectral basis that can be applied across all sequences regardless of content. Second, unlike models such as SR-GNN [2] that dynamically construct item-level graphs for each session—incurring significant computational overhead due to repeated graph building and message passing—TV-Rec operates on a fixed graph structure defined solely by sequence length. This enables highly efficient batched processing without any need for per-instance graph construction.
>
> ## **2. Comparison to graph-based methods**
> Our initial focus was on convolutional and hybrid baselines, as TV-Rec was designed to replace both self-attention and fixed convolution filters. However, we acknowledge that TV-Rec is grounded in graph signal processing principles, and thus comparisons with GNN-based sequential recommendation models are relevant. To address this, we have conducted additional experiments with four GNN-based baselines specifically designed for sequential recommendation (SR-GNN [2], GC-SAN [3], GCL4SR [4] and LS4SRec [5]) under the same settings as in Section 4.1. We note that most GCN-based recommendation models are designed for collaborative filtering with static user-item graphs, making them difficult to compare directly with sequential models like TV-Rec.
>
> ### **2-1. Performance Comparison**
>
>
> As shown in Table A, TV-Rec consistently outperforms these models across three datasets, confirming that its time-variant design captures temporal dependencies more effectively than message-passing GNN architectures. To ensure a comprehensive evaluation, we will report the results on the remaining three benchmark datasets during the discussion phase once the experiments are complete.
>
> **Table A. Comparison to GNN-based methods**
>
> | Datasets    | Metric    | SR-GNN  | GC-SAN  | GCL4SR | LS4SRec | TV-Rec |
> |-------------|-----------|---------|---------|--------|----------|---------|
> | **Beauty**  | Recall@5  | 0.0365  | 0.0570  | 0.0631 | 0.0626   | 0.0721  |
> |             | Recall@10 | 0.0567  | 0.0784  | 0.0880 | 0.0876   | 0.1017  |
> |             | Recall@20 | 0.0847  | 0.1059  | 0.1206 | 0.1200   | 0.1403  |
> |             | NDCG@5    | 0.0238  | 0.0408  | 0.0438 | 0.0442   | 0.0513  |
> |             | NDCG@10   | 0.0303  | 0.0477  | 0.0519 | 0.0522   | 0.0608  |
> |             | NDCG@20   | 0.0374  | 0.0546  | 0.0601 | 0.0604   | 0.0705  |
> | **LastFM**  | Recall@5  | 0.0385  | 0.0376  | 0.0385 | 0.0505   | 0.0596  |
> |             | Recall@10 | 0.0532  | 0.0569  | 0.0633 | 0.0752   | 0.0853  |
> |             | Recall@20 | 0.0872  | 0.0807  | 0.0908 | 0.1147   | 0.1202  |
> |             | NDCG@5    | 0.0248  | 0.0273  | 0.0249 | 0.0352   | 0.0402  |
> |             | NDCG@10   | 0.0295  | 0.0334  | 0.0328 | 0.0431   | 0.0484  |
> |             | NDCG@20   | 0.0379  | 0.0394  | 0.0398 | 0.0531   | 0.0572  |
> | **Foursquare** | Recall@5  | 0.0157  | 0.0148  | 0.0148 | 0.0148   | 0.0175  |
> |             | Recall@10 | 0.0203  | 0.0175  | 0.0166 | 0.0157   | 0.0259  |
> |             | Recall@20 | 0.0222  | 0.0212  | 0.0185 | 0.0194   | 0.0314  |
> |             | NDCG@5    | 0.0117  | 0.0114  | 0.0113 | 0.0124   | 0.0134  |
> |             | NDCG@10   | 0.0132  | 0.0122  | 0.0119 | 0.0127   | 0.0161  |
> |             | NDCG@20   | 0.0137  | 0.0131  | 0.0123 | 0.0136   | 0.0176  |
>
> ### **2-2. Architectural Comparison**
>
> SR-GNN and GC-SAN adopt local item transition graphs, where each session or user sequence is transformed into a directed graph based on item co-occurrence. On the other hand, GCL4SR and LS4SRec construct global item transition graphs (e.g., WITG) derived from the aggregated behavior of all users. While TV-Rec also leverages a graph-based view of sequential data, it differs from these baselines in several fundamental aspects.
>
> First, although the graph structure in TV-Rec is superficially similar to local item transition graphs, the key distinction lies in how the nodes are defined. Existing methods treat items as nodes, meaning that repeated occurrences of the same item in different positions map to the same node. In contrast, TV-Rec considers positions in the sequence as nodes, allowing even the same item to be treated as distinct nodes depending on its temporal position. This design choice enables the model to capture fine-grained temporal and positional dependencies, which are often overlooked when items are treated identically across time.
>
> Second, all four baselines explicitly employ GNNs to propagate and learn node representations. In contrast, TV-Rec does not use GNNs at all, and instead utilizes time-variant convolutional filters inspired by graph signal processing. These filters operate over a structured graph without relying on iterative message passing. As a result, TV-Rec achieves significantly faster training and inference times, while maintaining superior or comparable accuracy.
>
>
> ## **3. Statistical Significance Testing**
> To validate the robustness of TV-Rec, we conducted the Mann-Whitney U test [6], comparing its performance against the best baseline for each dataset using HR@20 and NDCG@20 metrics. The results show statistically significant improvements on all datasets, with p-values below 0.05 for each comparison:
>
> **Table B. Mann-Whitney U Test Result**
> | Dataset        | HR@20 p-value    | NDCG@20 p-value |
> |----------------|------------------|-----------------|
> | Beauty         | 5.31e-05         | 0.0329          |
> | Sports         | 1.38e-05         | 0.0200          |
> | Yelp           | 4.85e-05         | 0.0001          |
> | LastFM         | 5.53e-10         | 4.85e-06        |
> | Foursquare  | 7.74e-10         | 0.0388          |
> | ML-1M          | 0.0024           | 0.0205          |
>
> These results provide strong evidence that our TV-Rec consistently outperforms state-of-the-art (SOTA) baselines on all datasets, demonstrating its robustness across different metrics and datasets.
>
> ## **4. Regarding orthogonal regularization**
> Eq. (13), $B_{real}$ and $B_{imag}$ refer to the real and imaginary parts of the basis matrix $B$ used in constructing the filter tap matrix $H$ (see Eq. (8)). Since $B$ is complex-valued due to the use of GFT, we explicitly separate and regularize its real and imaginary components. Orthogonal regularization is applied to encourage the rows of $B$ to be orthonormal, which improves the numerical stability and expressiveness of the filter. This allows $B$ to span a well-conditioned subspace, reducing redundancy and improving generalization [7,8].
>
> ## **5. Regarding the experimental setting of basic graph filter in ablation study**
> We apologize for the lack of clarity in the ablation study. The basic graph filter in Section 4.3 refers to a version of TV-Rec where the time-variant filter is replaced with a standard (non-time-variant) graph filter—i.e., the latter case. This ablation shares the same overall architecture as TV-Rec, allowing for a direct comparison of filter designs. The formulation of this standard filter corresponds to Eq. (3). We will revise the caption to clarify this and avoid confusion.
>
>
> ## **Reference**
> [1] An Attentive Inductive Bias for Sequential Recommendation beyond Self-Attention, AAAI 2024
>
> [2] Session-Based Recommendation with Graph Neural Networks, AAAI 2019
>
> [3] Graph Contextualized Self-Attention Network for Session-based Recommendation, IJCAI 2019
>
> [4] Enhancing Sequential Recommendation with Graph Contrastive Learning, IJCAI 2022
>
> [5] Self-supervised Graph Neural Sequential Recommendation with Disentangling Long and Short-Term Interest, ACM Transactions on Recommender Systems 2025
>
> [6] "Wilcoxon-Mann-Whitney or t-test? On assumptions for hypothesis tests and multiple interpretations of decision rules." Statistics surveys 4 (2010).
>
> [7] Can We Gain More from Orthogonality Regularizations in Training Deep CNNs?, NeurIPS 2018
>
> [8] Orthogonal Graph Neural Networks, AAAI 2022

---

> > ### Comment · Reviewer_Y8Cz · 2025-08-04
> >
> > I thank the authors for addressing my previous questions and for their clarifications in response to my review. The additional comparison with graph-based baselines resolves most of my concerns, and I encourage the authors to include these results in the revised version. I have two follow-up questions:
> >
> > 1. In the comparison with graph-based methods, the authors report **Recall\@k**, whereas the original papers report **HR\@k**. Could the authors comment on this choice?
> >
> > 2. I examined the performance (HR\@k and NDCG\@k) of the best-performing baseline, BSARec, from its AAAI publication. I noticed that the results reported for BSARec in the authors’ manuscript are close to, but slightly lower than, those in the original paper. Could the authors clarify the reason for this performance gap? For instance, could this be due to differences in data splitting, random seeds, or other experimental settings?

---

> > > ### Author Response · Authors · 2025-08-06
> > >
> > > We sincerely thank the reviewer Y8Cz for the helpful follow-up questions and for pointing out important details. Please find our responses and updated experimental results below:
> > >
> > > ## **1. Recall@k vs HR@k**
> > >
> > > Thank you for pointing this out, and we apologize for the confusion. In sequential recommendation with a single ground-truth item, Recall@k and HR@k are equivalent. You may interpret the reported Recall@k as HR@k. We acknowledge the inconsistency in metric naming between the manuscript and the rebuttal and appreciate your attention to this point.
> > >
> > > ## **2. Performance gap with BSARec**
> > >
> > > Our implementation of BSARec is based on the official GitHub repository, and we used the same data splits and random seeds to ensure consistency. However, when applying the best hyperparameters reported in the BSARec paper to our environment, we were unable to reproduce the original performance. We suspect this discrepancy is due to differences in computing environments, including PyTorch versions, CUDA libraries, and GPU architectures, which are known to affect training dynamics and optimization behavior.
> > >
> > > ### Computing environments:
> > > * BSARec paper: Ubuntu 18.04 LTS, PyTorch 1.8.1, Intel i9 CPU, NVIDIA RTX 3090
> > > * Ours: Ubuntu 20.04 LTS, PyTorch 2.2.2, dual Intel Xeon CPUs, NVIDIA RTX A6000
> > >
> > > For your reference, we provide results obtained using the best hyperparameters reported in the BSARec paper under our environment. Note that the _tuned best hyperparams._ row corresponds to the results already presented in Table 2 of the main paper.
> > >
> > > **Table F. Performance of BSARec under our environment**
> > >
> > > | Dataset | Type                    | HR@5  | HR@10  | HR@20  | NDCG@05 | NDCG@10 | NDCG@20 |
> > > |---------|-------------------------|--------|--------|--------|---------|---------|---------|
> > > | **Beauty**  | paper best hyperparams.  | 0.0698 | 0.0980 | 0.1344 | 0.0495  | 0.0585  | 0.0677  |
> > > |         | tuned best hyperparams.  | 0.0714 | 0.0990 | 0.1393 | 0.0501  | 0.0590  | 0.0691  |
> > > | **Sports**  | paper best hyperparams.  | 0.0414 | 0.0595 | 0.0848 | 0.0288  | 0.0347  | 0.0410  |
> > > |         | tuned best hyperparams.  | 0.0422 | 0.0623 | 0.0865 | 0.0296  | 0.0361  | 0.0422  |
> > > | **Yelp**    | paper best hyperparams.  | 0.0254 | 0.0434 | 0.0694 | 0.0159  | 0.0217  | 0.0282  |
> > > |         | tuned best hyperparams.  | 0.0260 | 0.0446 | 0.0718 | 0.0162  | 0.0222  | 0.0290  |
> > > | **LastFM**  | paper best hyperparams.  | 0.0495 | 0.0670 | 0.1028 | 0.0335  | 0.0389  | 0.0480  |
> > > |         | tuned best hyperparams.  | 0.0505 | 0.0679 | 0.1119 | 0.0348  | 0.0405  | 0.0514  |
> > > | **ML-1M**   | paper best hyperparams.  | 0.1884 | 0.2745 | 0.3793 | 0.1254  | 0.1532  | 0.1795  |
> > > |         | tuned best hyperparams.  | 0.1909 | 0.2798 | 0.3844 | 0.1286  | 0.1573  | 0.1836  |

---

> > > ### Author Response · Authors · 2025-08-06
> > >
> > > ## **3. Performance Comparison to GNN-based methods**
> > >
> > > Thank you for your patience regarding the remaining results for the GNN-based baselines. We have now completed experiments on the additional benchmark datasets: Yelp, ML-1M, and Sports. Please refer to Table A (updated) for the complete results. As shown, TV-Rec consistently outperforms GNN-based methods across all six datasets, except for HR@10 on Yelp, further demonstrating the effectiveness of our time-variant filter design.
> > >
> > > **Table A (updated). Comparison to GNN-based methods**
> > >
> > > | Dataset | Metric   | SR-GNN | GC-SAN | GCL4SR | LS4SRec | TV-Rec |
> > > |--|----------|--------|--------|--------|---------|-|
> > > | **Beauty**  | HR@5  | 0.0365  | 0.0570  | 0.0631 | 0.0626   | 0.0721  |
> > > || HR@10 | 0.0567  | 0.0784  | 0.0880 | 0.0876   | 0.1017  |
> > > || HR@20 | 0.0847  | 0.1059  | 0.1206 | 0.1200   | 0.1403  |
> > > || NDCG@5    | 0.0238  | 0.0408  | 0.0438 | 0.0442   | 0.0513  |
> > > || NDCG@10   | 0.0303  | 0.0477  | 0.0519 | 0.0522   | 0.0608  |
> > > || NDCG@20   | 0.0374  | 0.0546  | 0.0601 | 0.0604   | 0.0705  |
> > > | **Sports**  | HR@5    | 0.0210 | 0.0295 | 0.0360 | 0.0353  | 0.0431 |
> > > || HR@10    | 0.0336 | 0.0423 | 0.0516 | 0.0516  | 0.0635 |
> > > |   | HR@20    | 0.0517 | 0.0608 | 0.0744 | 0.0758  | 0.0880 |
> > > |   | NDCG@05  | 0.0138 | 0.0201 | 0.0248 | 0.0244  | 0.0298 |
> > > |   | NDCG@10  | 0.0179 | 0.0242 | 0.0298 | 0.0296  | 0.0363 |
> > > |   | NDCG@20  | 0.0224 | 0.0289 | 0.0356 | 0.0357  | 0.0425 |
> > > | **Yelp**    | HR@5    | 0.0225 | 0.0236 | 0.0248 | 0.0278  |0.0290 |
> > > |    | HR@10    | 0.0377 | 0.0385 | 0.0417 | 0.0476  | 0.0474 |
> > > |    | HR@20    | 0.0609 | 0.0635 | 0.0684 | 0.0776  | 0.0777 |
> > > |    | NDCG@05  | 0.0145 | 0.0149 | 0.0155 | 0.0175  | 0.0186 |
> > > |    | NDCG@10  | 0.0194 | 0.0197 | 0.0209 | 0.0238  | 0.0245 |
> > > |    | NDCG@20  | 0.0252 | 0.0260 | 0.0276 | 0.0314  | 0.0321 |
> > > | **LastFM**  | HR@5  | 0.0385  | 0.0376  | 0.0385 | 0.0505   | 0.0596  |
> > > |             | HR@10 | 0.0532  | 0.0569  | 0.0633 | 0.0752   | 0.0853  |
> > > |             | HR@20 | 0.0872  | 0.0807  | 0.0908 | 0.1147   | 0.1202  |
> > > |             | NDCG@5    | 0.0248  | 0.0273  | 0.0249 | 0.0352   | 0.0402  |
> > > |             | NDCG@10   | 0.0295  | 0.0334  | 0.0328 | 0.0431   | 0.0484  |
> > > |             | NDCG@20   | 0.0379  | 0.0394  | 0.0398 | 0.0531   | 0.0572  |
> > > | **ML-1M**   | HR@5    | 0.1439 | 0.1724 | 0.1667 | 0.1758  | 0.2013 |
> > > |   | HR@10    | 0.2070 | 0.2449 | 0.2470 | 0.2579  | 0.2904 |
> > > |   | HR@20    | 0.2940 | 0.3255 | 0.3381 | 0.3675  | 0.4079 |
> > > |   | NDCG@05  | 0.0967 | 0.1173 | 0.1120 | 0.1195  | 0.1371 |
> > > |   | NDCG@10  | 0.1169 | 0.1407 | 0.1379 | 0.1459  | 0.1658 |
> > > |   | NDCG@20  | 0.1390 | 0.1611 | 0.1607 | 0.1735  | 0.1955 |
> > > | **Foursquare** | HR@5  | 0.0157  | 0.0148  | 0.0148 | 0.0148   | 0.0175  |
> > > |             | HR@10 | 0.0203  | 0.0175  | 0.0166 | 0.0157   | 0.0259  |
> > > |             | HR@20 | 0.0222  | 0.0212  | 0.0185 | 0.0194   | 0.0314  |
> > > |             | NDCG@5    | 0.0117  | 0.0114  | 0.0113 | 0.0124   | 0.0134  |
> > > |             | NDCG@10   | 0.0132  | 0.0122  | 0.0119 | 0.0127   | 0.0161  |
> > > |             | NDCG@20   | 0.0137  | 0.0131  | 0.0123 | 0.0136   | 0.0176  |

---

> > > > ### Comment · Reviewer_Y8Cz · 2025-08-08
> > > >
> > > > Thank you for the detailed rebuttal. I appreciate your inclusion of the additional comparison with the graph-based method, which addresses most of my earlier concerns. I encourage you to incorporate these results and clarifications into the final version of the paper. After also considering the other reviews, I did not find significant remaining weaknesses. I have accordingly increased my score.

---

### Official Review · Reviewer_HzEw · 2025-07-01

**Clarity:** 2
**Significance:** 2
**Originality:** 2
**Rating:** 4
**Confidence:** 3

**Summary:**

This article highlights the limitations of traditional convolutional filters in sequential recommendation tasks due to their fixed nature. To tackle this, the authors propose TV-Rec, a novel method that employs time-variant convolutional filters. These filters adaptively apply different filters to each data point, allowing for the capture of complex patterns. A key advantage of TV-Rec is its ability to achieve high performance in long-range modeling without the need for self-attention. Additionally, it boasts fast inference times due to its linear operator nature. The effectiveness and efficiency of TV-Rec have been rigorously validated through extensive experiments on six datasets.

**Questions:**

1. Given that TV-Rec is built on graph signal processing principles, why does the paper not include mainstream graph-based sequential recommendation models (e.g., those leveraging graph neural networks or graph convolutional networks) as baselines in its experimental evaluations?

2. Without comparisons to such graph-based models, how can readers assess whether TV-Rec offers significant advantages or uniqueness in capturing sequential dependencies compared to existing graph convolutional approaches?

3. The paper claims the time-variant graph filter is a core innovation, but does it include dedicated ablation studies that directly compare this filter with standard graph convolution operations under the same experimental settings to explicitly demonstrate the former’s superiority?  In the absence of such comparative ablation experiments, how can the claim that the time-variant graph filter is a more effective design choice than traditional graph convolution for modeling the dynamics and complexity of user sequences be validated?

**Ethical Concerns:**

["NO or VERY MINOR ethics concerns only"]

**Final Justification:**

The author's rebuttal has addressed most of my concerns. Thus, I will increase the score.

**Limitations:**

yes

**Quality:**

2

**Strengths And Weaknesses:**

TV-Rec addresses the inherent limitations of conventional fixed convolutional filters in super-resolution by introducing time-variant filters that dynamically adapt to each data point, enabling the capture of complex patterns without relying on self-attention mechanisms. This approach not only achieves high performance in long-range modeling but also maintains fast inference speeds due to its linear operator nature, as validated through extensive experiments across six datasets, demonstrating both superior effectiveness and practical efficiency.


A major drawback of the paper is its insufficient explanation of the practical implementation of the time-variant filter: while it cites graph signal processing concepts and provides mathematical formulations (e.g., filter matrix \(H\) and frequency-domain operations), the main text lacks a clear, step-by-step description of how the filter dynamically adapts per data point during inference, how the graph Fourier transforms interact with filter taps, or how parameters are learned during training. This omission hinders readability and reproducibility, particularly for readers unfamiliar with graph signal processing.

A critical omission in the paper is its failure to benchmark TV-Rec against existing graph-based sequential recommendation models (e.g., GNNs or graph convolutional approaches), despite its reliance on graph signal processing principles. This absence of comparison prevents readers from evaluating TV-Rec's relative performance and novelty within the broader graph-driven sequential recommendation literature, thereby undermining its contextual significance.

The paper suffers from overly terse table and figure captions (e.g., "Ablation studies" for Table 4 and "Visualization of learned graph filters on LastFM" for Figure 3), which omit critical context about experimental objectives, variables, and key insights. This lack of detail forces readers to deduce significance from the main text, diminishing the work's clarity and professional presentation.

The paper’s casual use of ambiguous abbreviations (e.g., "SR" for sequential recommendation without explicit definition) risks confusion, particularly for readers unfamiliar with the field. Moreover, Section 4.4 fails to provide proper citations for compared methods (AdaMCT, BSARec, SASRec, BERT4Rec, FMLPRec), hindering source verification and weakening the academic rigor of the complexity and runtime analysis.

---

> ### Author Rebuttal · Authors · 2025-07-31
>
> Thank you for your insightful and constructive feedback. In this response, we address each of your comments in detail and provide additional experimental results and clarifications.
>
> ## **1. Comparison with graph-based methods**
> Our initial focus was on convolutional and hybrid baselines, as TV-Rec was designed to replace both self-attention and fixed convolution filters. However, we acknowledge that TV-Rec is grounded in graph signal processing principles, and thus comparisons with GNN-based sequential recommendation models are relevant. To address this, we have conducted additional experiments with four GNN-based baselines specifically designed for sequential recommendation (SR-GNN [1], GC-SAN [2], GCL4SR [3] and LS4SRec [4]) under the same settings as in Section 4.1. We note that most GCN-based recommendation models are designed for collaborative filtering with static user-item graphs, making them difficult to compare directly with sequential models like TV-Rec.
>
> ### **1-1. Performance Comparison**
>
>
> As shown in Table A, TV-Rec consistently outperforms these models across three datasets, confirming that its time-variant design captures temporal dependencies more effectively than message-passing GNN architectures. To ensure a comprehensive evaluation, we will report the results on the remaining three benchmark datasets during the discussion phase once the experiments are complete.
>
> **Table A. Comparison to GNN-based methods**
>
> | Datasets    | Metric    | SR-GNN  | GC-SAN  | GCL4SR | LS4SRec | TV-Rec |
> |-------------|-----------|---------|---------|--------|----------|---------|
> | **Beauty**  | Recall@5  | 0.0365  | 0.0570  | 0.0631 | 0.0626   | 0.0721  |
> |             | Recall@10 | 0.0567  | 0.0784  | 0.0880 | 0.0876   | 0.1017  |
> |             | Recall@20 | 0.0847  | 0.1059  | 0.1206 | 0.1200   | 0.1403  |
> |             | NDCG@5    | 0.0238  | 0.0408  | 0.0438 | 0.0442   | 0.0513  |
> |             | NDCG@10   | 0.0303  | 0.0477  | 0.0519 | 0.0522   | 0.0608  |
> |             | NDCG@20   | 0.0374  | 0.0546  | 0.0601 | 0.0604   | 0.0705  |
> | **LastFM**  | Recall@5  | 0.0385  | 0.0376  | 0.0385 | 0.0505   | 0.0596  |
> |             | Recall@10 | 0.0532  | 0.0569  | 0.0633 | 0.0752   | 0.0853  |
> |             | Recall@20 | 0.0872  | 0.0807  | 0.0908 | 0.1147   | 0.1202  |
> |             | NDCG@5    | 0.0248  | 0.0273  | 0.0249 | 0.0352   | 0.0402  |
> |             | NDCG@10   | 0.0295  | 0.0334  | 0.0328 | 0.0431   | 0.0484  |
> |             | NDCG@20   | 0.0379  | 0.0394  | 0.0398 | 0.0531   | 0.0572  |
> | **Foursquare** | Recall@5  | 0.0157  | 0.0148  | 0.0148 | 0.0148   | 0.0175  |
> |             | Recall@10 | 0.0203  | 0.0175  | 0.0166 | 0.0157   | 0.0259  |
> |             | Recall@20 | 0.0222  | 0.0212  | 0.0185 | 0.0194   | 0.0314  |
> |             | NDCG@5    | 0.0117  | 0.0114  | 0.0113 | 0.0124   | 0.0134  |
> |             | NDCG@10   | 0.0132  | 0.0122  | 0.0119 | 0.0127   | 0.0161  |
> |             | NDCG@20   | 0.0137  | 0.0131  | 0.0123 | 0.0136   | 0.0176  |
>
> ### **1-2. Architectural Comparison**
>
> SR-GNN and GC-SAN adopt local item transition graphs, where each session or user sequence is transformed into a directed graph. On the other hand, GCL4SR and LS4SRec construct global item transition graphs (e.g., WITG) derived from the aggregated behavior of all users. While TV-Rec also leverages a graph-based view of sequential data, it differs from these baselines in several fundamental aspects.
>
> First, although the graph structure in TV-Rec is superficially similar to local item transition graphs, the key distinction lies in how the nodes are defined. Existing methods treat items as nodes, meaning that repeated occurrences of the same item in different positions map to the same node. In contrast, TV-Rec considers positions in the sequence as nodes, allowing even the same item to be treated as distinct nodes depending on its temporal position. This design choice enables the model to capture fine-grained temporal and positional dependencies, which are often overlooked when items are treated identically across time.
>
> Second, all four baselines explicitly employ GNNs to propagate and learn node representations. In contrast, TV-Rec does not use GNNs at all, and instead utilizes time-variant convolutional filters inspired by graph signal processing. These filters operate over a structured graph without relying on iterative message passing. As a result, TV-Rec achieves significantly faster training and inference times, while maintaining superior or comparable accuracy.
>
> ## **2. Time-variant graph filter vs. standard graph convolution**
> We would like to clarify that we already include an ablation study comparing our time-variant graph filter with a standard graph convolution (denoted as “Basic Graph Filter”) under identical conditions (Table 4, Section 4.3). As shown, replacing the time-variant filter with a standard graph filter leads to performance degradation across all datasets, clearly demonstrating the benefit of our design. To improve clarity, we agree that the caption or description in this section could better highlight this intent. We will revise it accordingly so that readers can more easily recognize the purpose and significance of this comparison.
>
> ## **3. Clarification on model behavior and filter design**
>
> We appreciate detailed feedback regarding the lack of practical implementation details for the time-variant filter. Below, we address the three key concerns raised: (1) the dynamic behavior during inference, (2) the interaction between the graph Fourier transform and filter taps, and (3) the parameter learning process.
> We agree that clearer explanations would improve readability and reproducibility, and we will incorporate these clarifications into the final version of the paper.
>
> ### **3-1. Dynamic behavior during inference**
> As noted in Appendix I, the term “time-variant” refers to the use of different filters across sequence positions—not to input-dependent adaptation. TV-Rec applies the same learned filters to all inputs during inference. These filters are position-specific but fixed once trained, making the model data-independent at test time and contributing to its computational efficiency. We will revise the text to avoid this confusion.
>
> ### **3-2. Interaction between GFT and filter taps**
> To clarify how the graph Fourier transform interacts with the learned filter taps, we provide the following pseudocode summarizing the spectral filtering process:
>
> ```python
> # X       : [batch_size, N, d]        # Input sequence
> # U       : [N, N]                    # GFT matrix
> # C       : [N, m]                    # Coefficient matrix
> # B       : [m, K+1]                  # Pre-normalized basis
> # Λ       : [N, K+1]                  # Vandermonde matrix (Λ)
>
> # Step 1: Transform input to spectral domain
> X_freq = U.T @ X                     # Graph Fourier Transform
>
> # Step 2: Construct spectral filter
> H = C @ B                            # [N, K+1], B is pre-normalized
> H_proj = H @ Λ.T                     # [N, N]
>
> # Step 3: Apply element-wise scaled inverse GFT
> F = U * H_proj                       # Element-wise product
>
> # Step 4: Apply filter and return to time domain
> X_out = F @ X_freq                   # Final output
> ```
> This filtering mechanism allows the learned filter taps to modulate graph frequency responses in a position-aware yet data-independent manner.
>
> ### **3-3. Parameter learning**
> All filter parameters—including the coefficient matrix $C$ and the basis matrix $B$—are learned jointly through standard backpropagation. To encourage diversity and improve numerical stability, we apply an orthogonality regularization to B (Eq. 13), promoting a well-conditioned and expressive spectral basis.
>
>
> ## **4. Clarifications on captions, terminology, and citations**
> We apologize for the lack of clarity in certain parts of the paper. We will revise the table and figure captions to provide clear and sufficient context. Additionally, we will include proper references for the baseline models used in the experiments.
> Regarding the terminology, we would like to point out that abbreviations such as "SR" are defined at the beginning of the Introduction. Nevertheless, we acknowledge the importance of clarity for a broader readers and will ensure that key terms are clearly introduced when they first appear in the main text.
>
> ## **Reference**
> [1] Session-Based Recommendation with Graph Neural Networks, AAAI 2019
>
> [2] Graph Contextualized Self-Attention Network for Session-based Recommendation, IJCAI 2019
>
> [3] Enhancing Sequential Recommendation with Graph Contrastive Learning, IJCAI 2022
>
> [4] Self-supervised Graph Neural Sequential Recommendation with Disentangling Long and Short-Term Interest, ACM Transactions on Recommender Systems 2025

---

> > ### Author Response · Authors · 2025-08-06
> >
> > Thank you for your patience regarding the remaining results for the GNN-based baselines. We have now completed experiments on the additional benchmark datasets: Yelp, ML-1M, and Sports. Please refer to Table A (updated) for the complete results. As shown, TV-Rec consistently outperforms GNN-based methods across all six datasets, except for HR@10 on Yelp, further demonstrating the effectiveness of our time-variant filter design.
> >
> > **Table A (updated). Comparison to GNN-based methods**
> >
> > | Dataset | Metric   | SR-GNN | GC-SAN | GCL4SR | LS4SRec | TV-Rec |
> > |--|----------|--------|--------|--------|---------|-|
> > | **Beauty**  | HR@5  | 0.0365  | 0.0570  | 0.0631 | 0.0626   | 0.0721  |
> > || HR@10 | 0.0567  | 0.0784  | 0.0880 | 0.0876   | 0.1017  |
> > || HR@20 | 0.0847  | 0.1059  | 0.1206 | 0.1200   | 0.1403  |
> > || NDCG@5    | 0.0238  | 0.0408  | 0.0438 | 0.0442   | 0.0513  |
> > || NDCG@10   | 0.0303  | 0.0477  | 0.0519 | 0.0522   | 0.0608  |
> > || NDCG@20   | 0.0374  | 0.0546  | 0.0601 | 0.0604   | 0.0705  |
> > | **Sports**  | HR@5    | 0.0210 | 0.0295 | 0.0360 | 0.0353  | 0.0431 |
> > || HR@10    | 0.0336 | 0.0423 | 0.0516 | 0.0516  | 0.0635 |
> > |   | HR@20    | 0.0517 | 0.0608 | 0.0744 | 0.0758  | 0.0880 |
> > |   | NDCG@05  | 0.0138 | 0.0201 | 0.0248 | 0.0244  | 0.0298 |
> > |   | NDCG@10  | 0.0179 | 0.0242 | 0.0298 | 0.0296  | 0.0363 |
> > |   | NDCG@20  | 0.0224 | 0.0289 | 0.0356 | 0.0357  | 0.0425 |
> > | **Yelp**    | HR@5    | 0.0225 | 0.0236 | 0.0248 | 0.0278  |0.0290 |
> > |    | HR@10    | 0.0377 | 0.0385 | 0.0417 | 0.0476  | 0.0474 |
> > |    | HR@20    | 0.0609 | 0.0635 | 0.0684 | 0.0776  | 0.0777 |
> > |    | NDCG@05  | 0.0145 | 0.0149 | 0.0155 | 0.0175  | 0.0186 |
> > |    | NDCG@10  | 0.0194 | 0.0197 | 0.0209 | 0.0238  | 0.0245 |
> > |    | NDCG@20  | 0.0252 | 0.0260 | 0.0276 | 0.0314  | 0.0321 |
> > | **LastFM**  | HR@5  | 0.0385  | 0.0376  | 0.0385 | 0.0505   | 0.0596  |
> > |             | HR@10 | 0.0532  | 0.0569  | 0.0633 | 0.0752   | 0.0853  |
> > |             | HR@20 | 0.0872  | 0.0807  | 0.0908 | 0.1147   | 0.1202  |
> > |             | NDCG@5    | 0.0248  | 0.0273  | 0.0249 | 0.0352   | 0.0402  |
> > |             | NDCG@10   | 0.0295  | 0.0334  | 0.0328 | 0.0431   | 0.0484  |
> > |             | NDCG@20   | 0.0379  | 0.0394  | 0.0398 | 0.0531   | 0.0572  |
> > | **ML-1M**   | HR@5    | 0.1439 | 0.1724 | 0.1667 | 0.1758  | 0.2013 |
> > |   | HR@10    | 0.2070 | 0.2449 | 0.2470 | 0.2579  | 0.2904 |
> > |   | HR@20    | 0.2940 | 0.3255 | 0.3381 | 0.3675  | 0.4079 |
> > |   | NDCG@05  | 0.0967 | 0.1173 | 0.1120 | 0.1195  | 0.1371 |
> > |   | NDCG@10  | 0.1169 | 0.1407 | 0.1379 | 0.1459  | 0.1658 |
> > |   | NDCG@20  | 0.1390 | 0.1611 | 0.1607 | 0.1735  | 0.1955 |
> > | **Foursquare** | HR@5  | 0.0157  | 0.0148  | 0.0148 | 0.0148   | 0.0175  |
> > |             | HR@10 | 0.0203  | 0.0175  | 0.0166 | 0.0157   | 0.0259  |
> > |             | HR@20 | 0.0222  | 0.0212  | 0.0185 | 0.0194   | 0.0314  |
> > |             | NDCG@5    | 0.0117  | 0.0114  | 0.0113 | 0.0124   | 0.0134  |
> > |             | NDCG@10   | 0.0132  | 0.0122  | 0.0119 | 0.0127   | 0.0161  |
> > |             | NDCG@20   | 0.0137  | 0.0131  | 0.0123 | 0.0136   | 0.0176  |

---

> > ### Author Response · Authors · 2025-08-09
> >
> > Dear Reviewer,
> >
> >
> > With only a few hours left in the reviewer–author discussion period, we would be grateful if you could review our rebuttal and share your thoughts. We have carefully addressed all of your comments and supported our clarifications with additional experiments. Your feedback at this final stage would be greatly appreciated.
> >
> >
> > Best regards,
> > Authors

---

### Note · Authors · 2025-08-12

Dear ACs and Reviewers,

We sincerely appreciate the constructive feedback from all reviewers. Based on your valuable comments, we have made the following improvements:

1. Extended experiments with recent and relevant baselines
    * Added 4 GNN-based models (SR-GNN, GC-SAN, GCL4SR, LS4SRec).
    * Added 1 recent model Tim4Rec.
    * Demonstrated consistent improvements over all baselines across 6 datasets.
2. Demonstrated improvements on large datasets
    * Added experiment results on the XLong dataset (69K users, 2.1M items, 66.8M interactions, avg. length 958.8).
    * Showed substantial gains over baselines, confirming scalability to sequences up to length 1,000.
3. Provided further architectural and design clarifications
    * Justified the use of the line-graph structure to ensure a consistent spectral basis for all sequences, enabling efficient batching without per-instance graph construction.
4. Addressed sparsity and personalization concerns
    * Conducted user/item sparsity experiments, grouping by interaction counts, demonstrating TV-Rec's robustness to data sparsity.
    * Suggested personalization strategies that can be incorporated without significantly increasing computation.

After our rebuttal, 2 reviewers expressed satisfaction with our responses:

* Reviewer Y8Cz updated their recommendation from _borderline accept_ to _accept_, stating that they "did not find significant remaining weaknesses."
* Reviewer ANZm found the rebuttal acceptable and maintained their positive score.

However, reviewer HzEw has not yet responded to our initial rebuttal, and reviewer xQNR has not yet responded to our additional rebuttal. Nevertheless, we believe our responses have fully addressed their concerns, including additional GNN-based comparisons, detailed filter implementation explanations, and new results clarifying personalization and sparsity.

Again, we would like to thank the ACs and all the reviewers for your great efforts in reviewing our paper.

Best regards,

TV-Rec Authors

---

### Decision · Program_Chairs · 2025-09-17

**Decision:**

Accept (poster)

**Comment:**

This paper introduces TV-Rec, a novel sequential recommendation model leveraging time-variant convolutional filters inspired by graph signal processing. By dynamically adapting filters to each temporal position, TV-Rec captures complex sequential dependencies without relying on self-attention or positional embeddings, enabling both high performance and efficient inference. The approach is well-motivated, with a clear low-rank formulation and spectral filter design, offering interpretability and a compelling alternative to Transformer-based methods. Extensive experiments demonstrate consistent improvements over ten state-of-the-art models, supported by ablation studies, efficiency analyses, and qualitative visualizations. Some parts, such as sensitivity to hyperparameters, heuristic choices in the basis construction, and comparisons to GNN-based sequential recommenders, could be further improved.